# Cellular localization of the cell cycle inhibitor Cdkn1c controls growth arrest of adult skeletal muscle stem cells

Despoina Mademtzoglou[1,2], Yoko Asakura[3], Matthew J Borok[1,2], Sonia Alonso-Martin[1,2†], Philippos Mourikis[1,2], Yusaku Kodaka[3], Amrudha Mohan[3], Atsushi Asakura[3]*, Frederic Relaix[1,2,4,5]*

[1]Inserm, IMRB U955-E10, F-94010, Créteil, France; [2]Ecole Nationale Veterinaire d'Alfort, Faculté de medecine, F-94000, Université Paris-Est Creteil, Maison Alfort, France; [3]Stem Cell Institute, Paul and Sheila Wellstone Muscular Dystrophy Center, Department of Neurology, University of Minnesota Medical School, Minneapolis, United States; [4]Etablissement Français du Sang, Créteil, France; [5]APHP, Hopitaux Universitaires Henri Mondor, DHU Pepsy & Centre de Référence des Maladies Neuromusculaires GNMH, Créteil, France

**Abstract** Adult skeletal muscle maintenance and regeneration depend on efficient muscle stem cell (MuSC) functions. The mechanisms coordinating cell cycle with activation, renewal, and differentiation of MuSCs remain poorly understood. Here, we investigated how adult MuSCs are regulated by CDKN1c (p57[kip2]), a cyclin-dependent kinase inhibitor, using mouse molecular genetics. In the absence of CDKN1c, skeletal muscle repair is severely impaired after injury. We show that CDKN1c is not expressed in quiescent MuSCs, while being induced in activated and proliferating myoblasts and maintained in differentiating myogenic cells. In agreement, isolated *Cdkn1c*-deficient primary myoblasts display differentiation defects and increased proliferation. We further show that the subcellular localization of CDKN1c is dynamic; while CDKN1c is initially localized to the cytoplasm of activated/proliferating myoblasts, progressive nuclear translocation leads to growth arrest during differentiation. We propose that CDKN1c activity is restricted to differentiating myoblasts by regulated cyto-nuclear relocalization, coordinating the balance between proliferation and growth arrest.
DOI: https://doi.org/10.7554/eLife.33337.001

*For correspondence:
asakura@umn.edu (AA);
frelaix@gmail.com (FR)

Present address: †Tissue
Regeneration Laboratory, Centro
Nacional de Investigaciones
Cardiovasculares (CNIC), Madrid,
Spain

Competing interests: The
authors declare that no
competing interests exist.

Reviewing editor: Vittorio
Sartorelli, National Institute of
Arthritis, Musculoskeletal and
Skin Diseases, National Institutes
of Health, United States

## Introduction

Tissue regeneration is of vital importance for restoring tissue structure and function following damage. Skeletal muscle has a remarkable capacity to self-repair after severe injuries, a process dependent on muscle stem cells (MuSCs) (*Relaix and Zammit, 2012*). MuSCs originate from a PAX3/7+ progenitor cells that in late fetal life acquire their characteristic anatomical position between the basal lamina and the plasma membrane of muscle fibers (*Mauro, 1961*; *Relaix et al., 2005*). MuSCs are indispensable for postnatal muscle growth (*White et al., 2010*; *Pawlikowski et al., 2015*), maintenance (*Keefe et al., 2015*; *Pawlikowski et al., 2015*), and regeneration upon injury (*Lepper et al., 2011*; *McCarthy et al., 2011*; *Murphy et al., 2011*; *Sambasivan et al., 2011*). The majority of juvenile MuSCs acquire a non-proliferative, quiescent state around 3 weeks of post-natal life to ensure cellular/genomic integrity and long-term survival (*Lepper et al., 2009*; *White et al., 2010*; *Cheung and Rando, 2013*). However, once stimulated by homeostatic demand or damage, adult MuSCs exit quiescence, re-enter the cell cycle, and provide differentiated progeny for muscle repair, while a subpopulation self-renews to preserve the quiescent pool (*Relaix and Zammit,*

*2012*). The myogenic regulatory factors including MYOD and MYOGENIN orchestrate MuSCs commitment and progression through the myogenic lineage (*Wang et al., 2014*), while the signals that trigger cell cycle exit and (re-) entry into quiescence remain more elusive.

Cell cycle is a tightly synchronized process responding to positive and negative signals, while inappropriate growth arrest can result in cancer, malformations during development, and defective stem cell renewal (*Zhang et al., 1997*; *Matsumoto et al., 2011*; *Sherr, 2012*). Cyclin-dependent kinase inhibitors (CDKIs) are negative cell cycle regulators. CDKIs are divided into two structurally and functionally defined families: the INK4 family (including Cdkn2a, Cdkn2b, Cdkn2c, and Cdkn2d) and the Cip/Kip family (including Cdkn1a, Cdkn1b, and Cdkn1c) (*Borriello et al., 2011*).

Although members of the Cip/Kip family have been shown to inhibit proliferation and promote differentiation in embryonic muscle or in myoblasts in vitro, their involvement in postnatal MuSC cell cycle regulation is less well documented (*Halevy et al., 1995*; *Reynaud et al., 1999*; *Zhang et al., 1999*; *Messina et al., 2005*; *Chakkalakal et al., 2014*; *Zalc et al., 2014*). Given the emerging importance of Cdkn1c in stem cell quiescence (*Matsumoto et al., 2011*; *Zou et al., 2011*; *Furutachi et al., 2013*), we explored its role in adult myogenesis and MuSC-supported regeneration. Using mouse mutants and ex vivo analysis, we provide evidence that Cdkn1c is involved in the early phase following MuSC activation events, with its genetic ablation leading to impaired muscle regeneration associated with decreased differentiation and increased proliferation. We further show that Cdkn1c is not detected in quiescent MuSCs but is induced upon activation and maintained in differentiating myogenic cells. Finally, we show that Cdkn1c subcellular localization is specifically regulated during adult myogenesis, with a progressive cytoplasmic to nuclear translocation as activated myoblasts proceed to differentiation. Our results suggest that muscle stem cells require Cdkn1c activity for the dynamic control of growth arrest during adult myogenesis.

## Results

### Cdkn1c is required for postnatal myogenesis

Although *Cdkn1c* loss is generally associated with perinatal death (*Yan et al., 1997*; *Zhang et al., 1997*; *Susaki et al., 2009*; *Mademtzoglou et al., 2017*), a few *Cdkn1c* mutant mice survived in a mixed CD1;B6 background (4.2%; *Figure 1—figure supplement 1A*). *Cdkn1c*-deficient mice displayed reduced body weight compared to control littermates (*Figure 1—figure supplement 1B–C*). However, there was no significant difference in forelimb grip strength between *Cdkn1c* mutant and control mice at 1 or 2 months of age when the strength was calculated on a per weight basis. Strength was even slightly higher in 3- or 4-month-old *Cdkn1c* mutant mice compared to controls (*Figure 1—figure supplement 1D*). This difference could be explained by the smaller body weight of *Cdkn1c* mutants, possibly leading to increased relative grip strength (N/kg) in mutants. To evaluate the role of CDKN1c in muscle homeostasis, we examined sections of the hindlimb *Tibialis anterior* (TA) muscles in adult mice. Histological analysis showed that *Cdkn1c* knock-out muscles contained smaller fibers and displayed increased fibrosis (*Figure 1A–D*), implying hindered myogenic differentiation. The amount of centrally located nuclei, indicative of ongoing regeneration, was comparable in mutants and controls (*Figure 1E*). Myofiber culture conditions used allow MuSCs to become activated, start dividing (T24-48), and eventually, proceed to myogenic differentiation or self-renewal of the quiescent pool (T72) (Zammit et al., 2004). The number of PAX7+ MuSC on freshly isolated myofibers of *Extensor digitorum longus* (EDLs) was increased in *Cdkn1c* mutant mice compared to the controls (*Figure 1F–G*). Furthermore, PAX7+ MuSCs on *Cdkn1c* mutant myofibers were mostly MYOD-, at a similar percentage to controls (*Figure 1H*), indicating that Cdkn1c is not regulating MuSCs quiescence. When single myofibers were cultured for 72 hr (T72), *Cdkn1c* mutants displayed an increased ratio of PAX7+ MYOD- self-renewing cells and a decreased ratio of PAX7- MYOD+ differentiating myoblasts (*Figure 1I–J*). Taken together, our data suggest that in the absence of CDKN1c the MuSC compartment is correctly established, but a proportion of the MuSC population undergo increased self-renewal at the expense of differentiation.

Next, we evaluated the impact of CDKN1c loss on skeletal muscle regeneration. We performed intramuscular cardiotoxin (CTX) injections into the *Tibialis anterior* (TA) and sacrificed the mice at 3, (d3), 4 (d4), 7 (d7), and thirty (d30) days post-injury, to evaluate early and late time points of the regeneration procedure. Once muscle degeneration is induced, MuSCs undergo: (1) activation, (2)

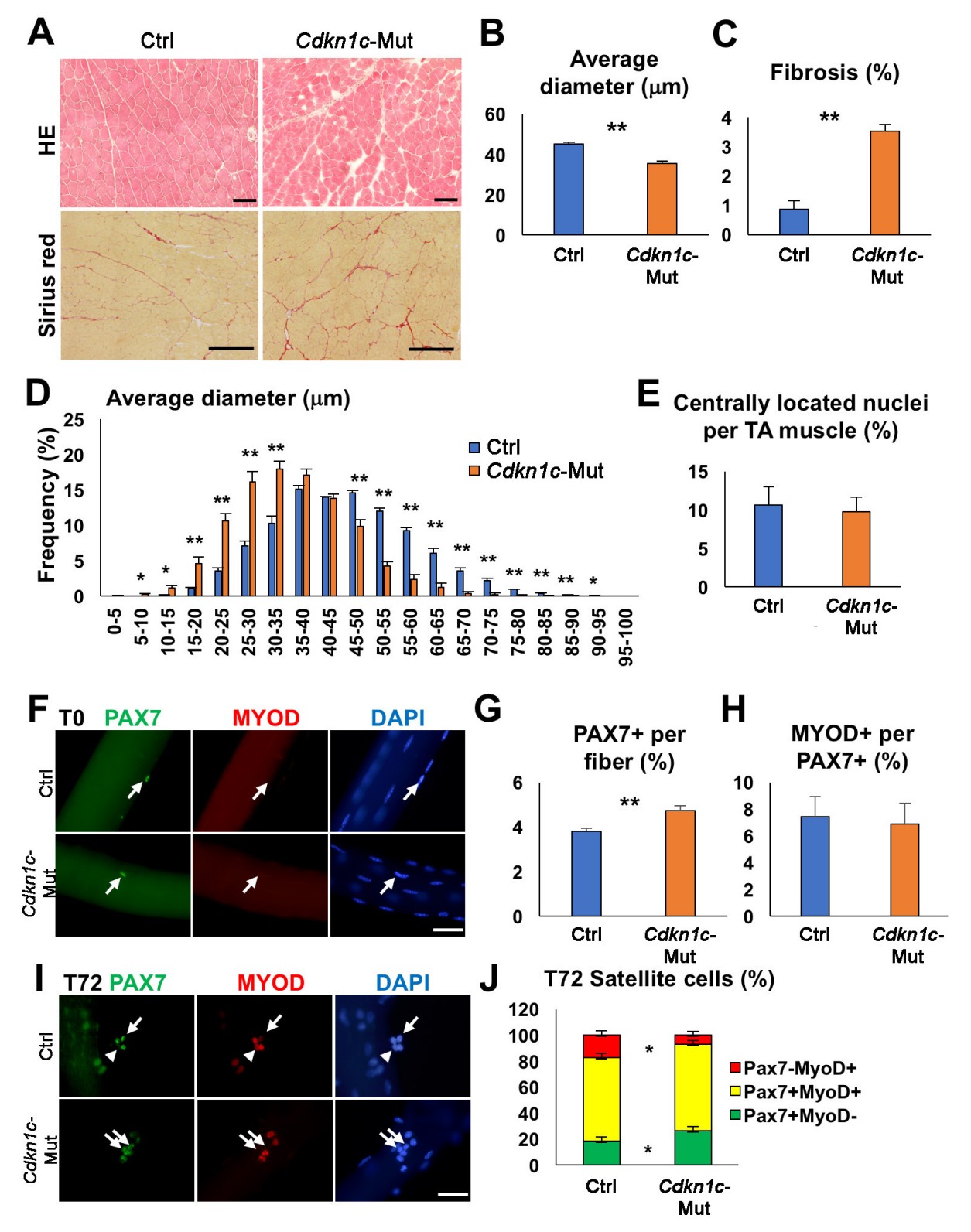

**Figure 1.** *Cdkn1c* deficiency impairs normal muscle growth. (A) Hematoxylin and Eosin (HE) and Sirius red staining of control (Ctrl) and *Cdkn1c* mutant (*Cdkn1c*-Mut) mouse *Tibialis anterior* (TA) muscles were performed to examine muscle histology, centrally located nucleated myofibers, and fibrosis. Scale bars, 100 μm. (B) Histogram showing the average of myofiber diameters (μm). (C) Histogram of average fibrotic area per TA muscle. (D) Fiber size (μm) distribution in control and *Cdkn1c* mutant mice. (E) Histogram of number of fibers with centrally located nuclei. (F) PAX7+ (green) MuSCs (arrows)
*Figure 1 continued on next page*

*Figure 1 continued*

on the myofibers isolated from EDL muscles of *Cdkn1c* mutant and control mice. MYOD (red) is not normally expressed in PAX7+ MuSCs at T0 (quiescence). DAPI (blue) shows all nuclei. Scale bars, 50 µm. (**G**) Numbers of PAX7+ satellite cells on the myofibers isolated from EDL. (**H**) Ratio of MYOD+ activated cells per PAX7+ MuSC on the myofibers isolated from EDL muscles of *Cdkn1c* mutant and control mice. (**I**) Immunofluorescence for PAX7 (green) and MYOD (red) at T72 in single myofiber cultures. Arrows and arrowheads show PAX7+MYOD- quiescent satellite cells and PAX7-MYOD+ differentiating cells, respectively. Scale bars, 50 µm. (**J**) Quantification of ratios of PAX7+ and MYOD+ cells per fiber at T72. Nuclei were counterstained with DAPI. *p≤0.05, **p≤0.01.

DOI: https://doi.org/10.7554/eLife.33337.002

The following figure supplement is available for figure 1:

**Figure supplement 1.** *Cdkn1c* mutant mice display smaller body weight.

DOI: https://doi.org/10.7554/eLife.33337.003

proliferation to expand their population, (3) self-renewal of the quiescent pool for future needs, and (4) differentiation for newly generated fibers and muscle repair (*Relaix and Zammit, 2012*). At d3 post-injury, loss of *Cdkn1c* promoted myoblasts proliferation and counteracted differentiation, as shown by increased EdU+ incorporation and reduced MYOD+EdU+ fraction, respectively. (*Figure 2—figure supplement 1A,B*). At d4 post-injury, *Cdkn1c*-deficient muscles showed smaller embryonic myosin (eMyHC)+ myofibers, a marker for early myofiber formation, compared to controls (*Figure 2A,C*). At d7 post-injury, *Cdkn1c*-deficient muscles showed increased cell infiltration and smaller and heterogeneous myofiber formation (*Figure 2A,B,D*), suggesting a delay in the regeneration process. At d30, signs of impaired regeneration associated with smaller fibers were observed (*Figure 2A,B,E*), including deposition of fibrotic tissue (*Figure 2A,F*) in *Cdkn1c*-mutant mice. Next, we evaluated the MuSC population from isolated myofibers of EDLs at d30 and TA muscle sections. The number of PAX7+ MuSCs was increased in *Cdkn1c* mutants compared to the controls (*Figure 2G–H*; *Figure 2—figure supplement 1C,D*) while the proportion of MYOD+ MuSC was not altered (*Figure 2I*). Therefore, our data suggest that Cdkn1c is required for postnatal muscle repair. In addition, *Cdkn1c* mutant myogenic cells demonstrated increased propensity for stem-cell self-renewal during both tissue establishment and regeneration.

## CDKN1c regulates the balance between proliferation and differentiation in activated myoblasts

We next abrogated CDKN1c specifically in adult MuSC to avoid cell non-autonomous effects and bypass the impact of CDKN1c loss during development. To conditionally ablate *Cdkn1c*, we have generated a floxed *Cdkn1c* allele (*Cdkn1c^Flox^*) to enable conditional knock-out of *Cdkn1c* using the Cre/loxP system (*Mademtzoglou et al., 2017*). Given that *Cdkn1c* is an imprinted gene expressed only by the maternal allele (*Matsuoka et al., 1995*), we used heterozygotes with maternal inheritance of *Cdkn1c^Flox^*, hereafter indicated as *Cdkn1c^Flox(m)/+^*. To specifically target the MuSCs and their progeny, we intercrossed *Cdkn1c^Flox^* mice with the *Pax7^CreERT2^* line (*Lepper et al., 2009*). In the compound mice, tamoxifen (TMX) administration results in CreERT2 translocation to the nucleus and *Cdkn1c* excision in the MuSC lineage. One week of TMX administration led to 84.5% recombination efficiency (*Figure 3—figure supplement 1A–B*). We then inserted *Cdkn1c^Flox^* and *Pax7^CreERT2^* in the *ROSA^mTmG^* background (*Muzumdar et al., 2007*). These double-fluorescent mice report Cre activity by expressing membrane-Tomato (mT) prior to Cre-mediated excision and membrane-GFP (mG) after excision. This allowed us to selectively isolate recombined (GFP+) MuSC cells.

We isolated by flow cytometry GFP+ MuSCs of control (*Pax7^CreERT2^*; *Cdkn1c^+/+^*; *ROSA^mTmG/+^*) and *Cdkn1c*-deficient (*Pax7^CreERT2^*; *Cdkn1c^Flox(m)/+^*; *ROSA^mTmG/+^*) mice. Collected cells grew in expansion conditions (20% fetal bovine serum, 10% horse serum) up to 70% confluence and then were serum-deprived to differentiate for 1 or 3 days (*Figure 3A*). *Cdkn1c* transcript and protein were induced by differentiation in control cells (*Figure 3B–C*). In contrast, they were not detected in cells from mutant animals (*Figure 3B,D*), demonstrating efficient Cre-mediated recombination of the *Cdkn1c* allele in FACS-isolated cells. We then performed primary myoblast cultures from control and *Cdkn1c*-deficient mice. We monitored proliferating cells under 'growth' (high serum) or 'differentiation' (low serum) conditions, and at d1 and d3 post-differentiation we stained with early (e.g. MYOGENIN) or late (i.e. Myosin heavy chain, MyHC) differentiation markers, respectively. Consistent with the function of CDKN1c as a cell cycle inhibitor, we observed increased proliferation in primary

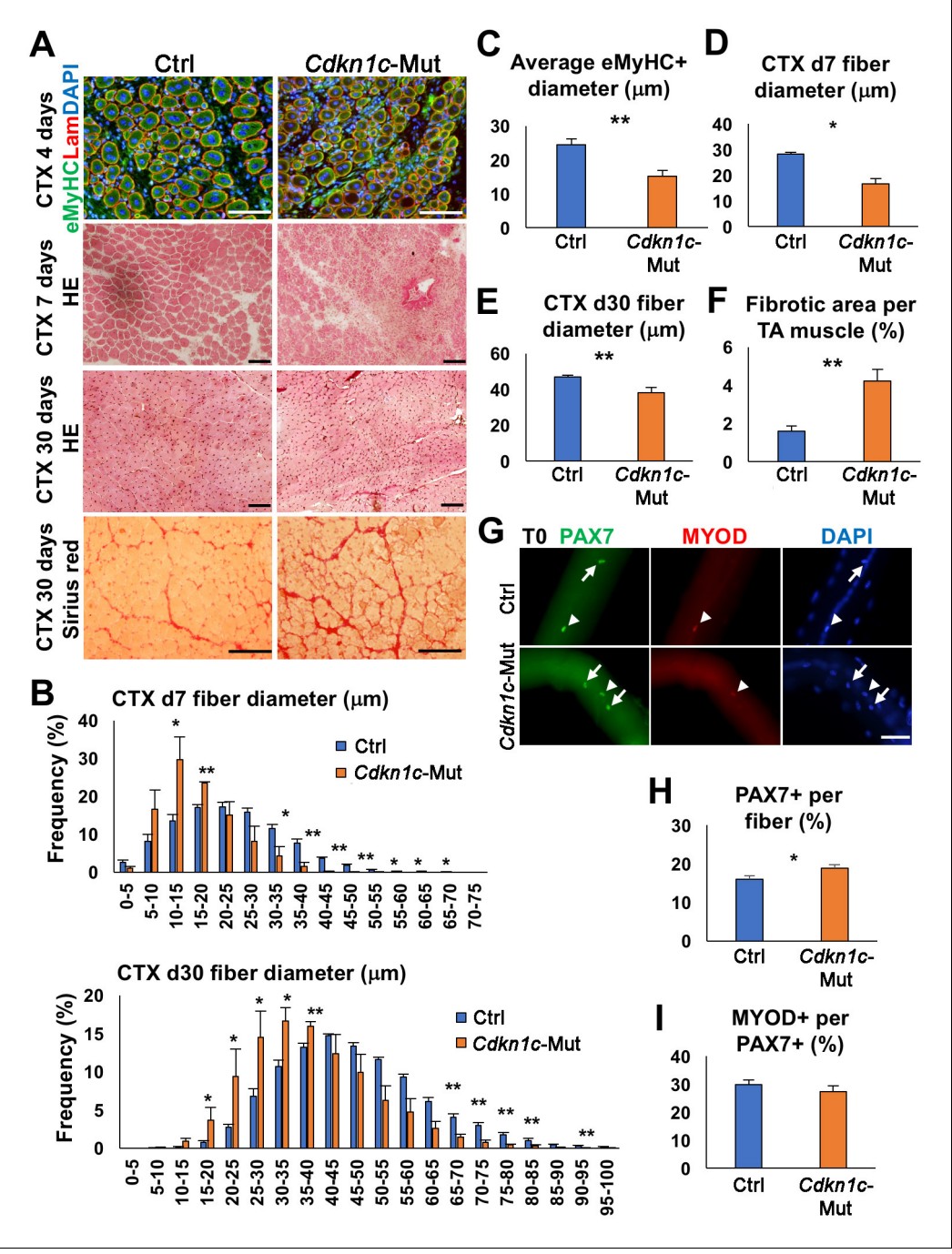

**Figure 2.** CDKN1c deficiency delays muscle regeneration. (**A**) Embryonic myosin (eMyHC)/LAMININ/DAPI, Hematoxylin and Eosin (HE), and Sirius red staining of twelve- to fifteen-week-old control (Ctrl) and *Cdkn1c* mutant mouse TA muscles were performed for histological and fibrosis characterization 4, 7 or thirty days after cardiotoxin (CTX) injection. Scale bars, 100 μm. (**B**) Fiber size (μm) distribution in control (Ctrl) and *Cdkn1c* mutant (*Cdkn1c-*Mut) mice 7 (upper panel) or thirty (lower panel) days after CTX injection. (**C**) Histogram of average embryonic MyHC+ fiber diameters (μm) 4 days after CTX injection. (**D**) Histogram of average fiber diameters (μm) 7 and thirty days after CTX injection. (**E**) Fiber size (μm) distribution in control and *Cdkn1c* mutant mice thirty days after CTX injection. (**F**) Histogram of average fibrotic area per TA muscle. (**G**) PAX7+ (green) MuSCs (arrows) on the myofibers isolated from EDL muscles of *Cdkn1c* mutant and control mice thirty days after CTX injection. MYOD (red) is occasionally expressed in PAX7+ MuSCs (arrow heads). DAPI (blue) shows all nuclei. Scale bars, 50 μm. (**H**) Numbers of PAX7+ MuSCs on the EDL isolated myofibers . (**I**) Ratio of MYOD+ activated cells per PAX7+ MuSC

*Figure 2 continued on next page*

*Figure 2 continued*

on the myofibers isolated from EDL muscles of *Cdkn1c* mutant and control mice. Nuclei were counter-stained with DAPI. Scale bars, 100 μm. *p≤0.05, **p≤0.01.

DOI: https://doi.org/10.7554/eLife.33337.004

The following figure supplement is available for figure 2:

**Figure supplement 1.** Myoblasts in *Cdkn1c* mutant mice display delayed cell cycle exit during muscle regeneration.

DOI: https://doi.org/10.7554/eLife.33337.005

myoblasts derived from MuSC-specific *Cdkn1c* mutant mice. When cells were maintained in growth conditions for 5 days, we observed 50% and 38% more EdU+ and KI67+ cells, respectively, in the absence of CDKN1c (*Figure 3E–H*), suggesting that CDKN1c is involved in restraining cell cycle progression. Slightly more PAX7+ EdU+ cells were observed in the *Cdkn1c*-deficient myoblast cultures, while differences in MYOD+ EdU+ or MYOGENIN+ EdU+ populations did now show significant differences (*Figure 3—figure supplement 1C–F*). Furthermore, in cultures of FACS-isolated MuSCs from MuSC-specific *Cdkn1c*-deficient mice, myogenic differentiation was impaired. One-day post-differentiation, MYOGENIN was significantly decreased (*Figure 3I,K,L*). In addition, myotube formation 3 days post-differentiation was severely compromised (*Figure 3J,M,N*). We also detected a similar increase in cell proliferation and reduction in myogenic differentiation in primary myoblasts isolated from global *Cdkn1c* mutant mice (*Figure 3—figure supplement 2*). In conclusion, both MuSC-specific and global ablation of CDKN1c led to increased primary myoblasts proliferation at the expense of myogenic differentiation. Together, our data demonstrate that CDKN1c is required in MuSCs for correct cell cycle regulation and differentiation during postnatal myogenesis.

## Satellite-cell-specific CDKN1c loss compromises muscle regeneration

We next evaluated the impact of MuSC-specific *Cdkn1c* ablation, driven by *Pax7*$^{CreERT2}$ (*Lepper et al., 2009*), on skeletal muscle regeneration after injury. Following 4 weeks of tamoxifen administration to induce recombination in adult (8- to 12-week-old) mice, we injected control and *Pax7*$^{CreERT2/+}$; *Cdkn1c*$^{Flox}$ (*Cdkn1c* cKO) TA muscles with CTX and evaluated muscle repair at 7 days post-injury (*Figure 4A*), having achieved 94.5% recombination efficiency in MuSCs (*Figure 4—figure supplement 1A–B*). Of note, *Pax7*$^{CreERT2/+}$ (hereafter, Cre control) animals only express one allele of *Pax7*, as the Cre recombinase is inserted into the *Pax7* gene. Hence, in all experiments we also included Cre control mice, as we found that *Pax7*$^{CreERT2}$ (*Lepper et al., 2009*) heterozygous mice display a mild regeneration phenotype (*Figure 4B–G'*), hence potentially acting as sensitizing background. Histological analysis at d7 post-injury showed normal tissue repair in wild-type (Wt) muscles (*Figure 4B,C*). In contrast, Cre control muscles showed increased cell infiltration and reduced regenerated myofibers of smaller sizes compared to Wt (*Figure 4B'*; *Figure 4—figure supplement 1C–D*), while muscle regeneration was severely compromised in *Cdkn1c* cKO muscles, in which newly formed myofibers were rare and small and most of the tissue consisted of cell infiltration and fat deposits (*Figure 4B'', C''*; *Figure 4—figure supplement 1C–D*). Furthermore, induction of early differentiation was delayed, as evidenced by the presence of embryonic eMyHC+ fibers in *Cdkn1c* cKO and Cre control, but not in Wt muscles at this time point (*Figure 4D–E''*). Consistent with the lack of tissue regeneration, we observed a massive diminution of the MuSC compartment in *Cdkn1c* cKO muscles (*Figure 4F–H*). Together, our data reveal that CDKN1c expression in the myogenic lineage is required for muscle repair.

## Dynamic CDKN1c expression in adult myogenesis

To analyze the expression of CDKN1c during MuSC-mediated myogenesis, we isolated single myofibers with their associated PAX7+ MuSCs from EDL muscles of adult wild-type mice. Immunostaining experiments, using a CDKN1c-specific antibody, showed that quiescent MuSCs were not labeled (*Figure 5—figure supplement 1A*). Similarly, MuSCs of resting adult TA muscles were CDKN1c-negative following labeling of cryosections (*Figure 5—figure supplement 1B*). To evaluate the kinetics of CDKN1c expression during activation, self-renewal, and differentiation, we analyzed myofibers that were cultured from 24 to 72 hr. *Zammit et al., 2004*During the initial proliferation state (T24-

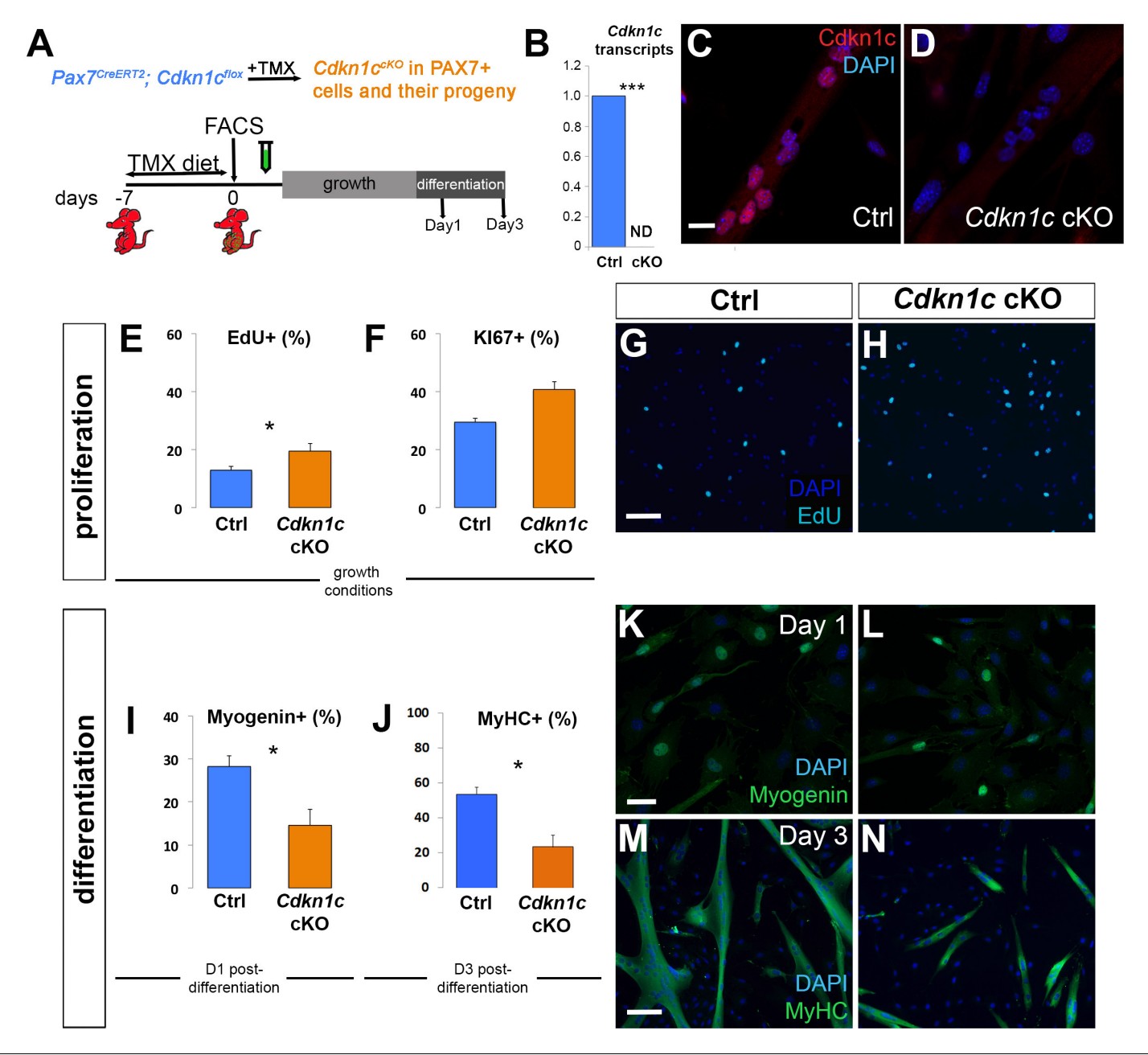

**Figure 3.** *Cdkn1c* deficiency impairs myogenic differentiation. (**A**) Time-course of tamoxifen (TMX) administration, muscle satellite cell harvest (FACS arrow) and culture (light gray bar for growth culture conditions, dark gray bar for differentiation culture conditions). Analyzed animals were *Pax7*CreERT2/+; *Cdkn1c*Flox(m)/+;*Rosa*mTmG (*Cdkn1c* cKO) and *Pax7*CreERT2/+; *Cdkn1c*+/+;*Rosa*mTmG (control; Ctrl); maternal inheritance of the imprinted *Cdkn1c* is indicated by superscript (m). (**B**) *Cdkn1c* transcript levels of control and *Cdkn1c* cKO myoblast cultures 3 days post-differentiation. ND; not detected. (**C–D**) Control (**C**) and *Cdkn1c* cKO (**D**) myoblast cultures were examined for CDKN1c protein (red) following three days under differentiation conditions. (**E–N**) Control and *Cdkn1c* cKO myoblast cultures were examined for EdU+ (light blue) cells (**E, G, H**), KI67+ cells (**F**), MYOGENIN+ cells (green; **I, K, L**), and myotube formation (**J, M, N**). Nascent myotubes were marked with myosin heavy chain (MyHC; green; **M, N**). Nuclei were counter-stained with DAPI (blue). Graphs show quantification of EdU and KI67 expression under growth conditions (**E, F**), MYOGENIN expression following 24 hr under differentiation conditions (**I**), and MyHC+ cells following 72 hr under differentiation conditions (**J**). Data show mean +SD, n = 3 animals. Asterisks indicate significance; *p≤0.05, ***p≤0.001. Scale bars, 40 μm (**C, K**), 1000 μm (**G, M**).
DOI: https://doi.org/10.7554/eLife.33337.006

The following figure supplements are available for figure 3:

**Figure supplement 1.** Myogenic marker expression in proliferating myoblasts.

*Figure 3 continued on next page*

*Figure 3 continued*

DOI: https://doi.org/10.7554/eLife.33337.007

**Figure supplement 2.** *Cdkn1c* mutant myoblasts display increased proliferation and reduced differentiation.

DOI: https://doi.org/10.7554/eLife.33337.008

T48) MuSCs co-express PAX7 and MYOD (*Zammit et al., 2004*). Activated PAX7+ and MYOD+

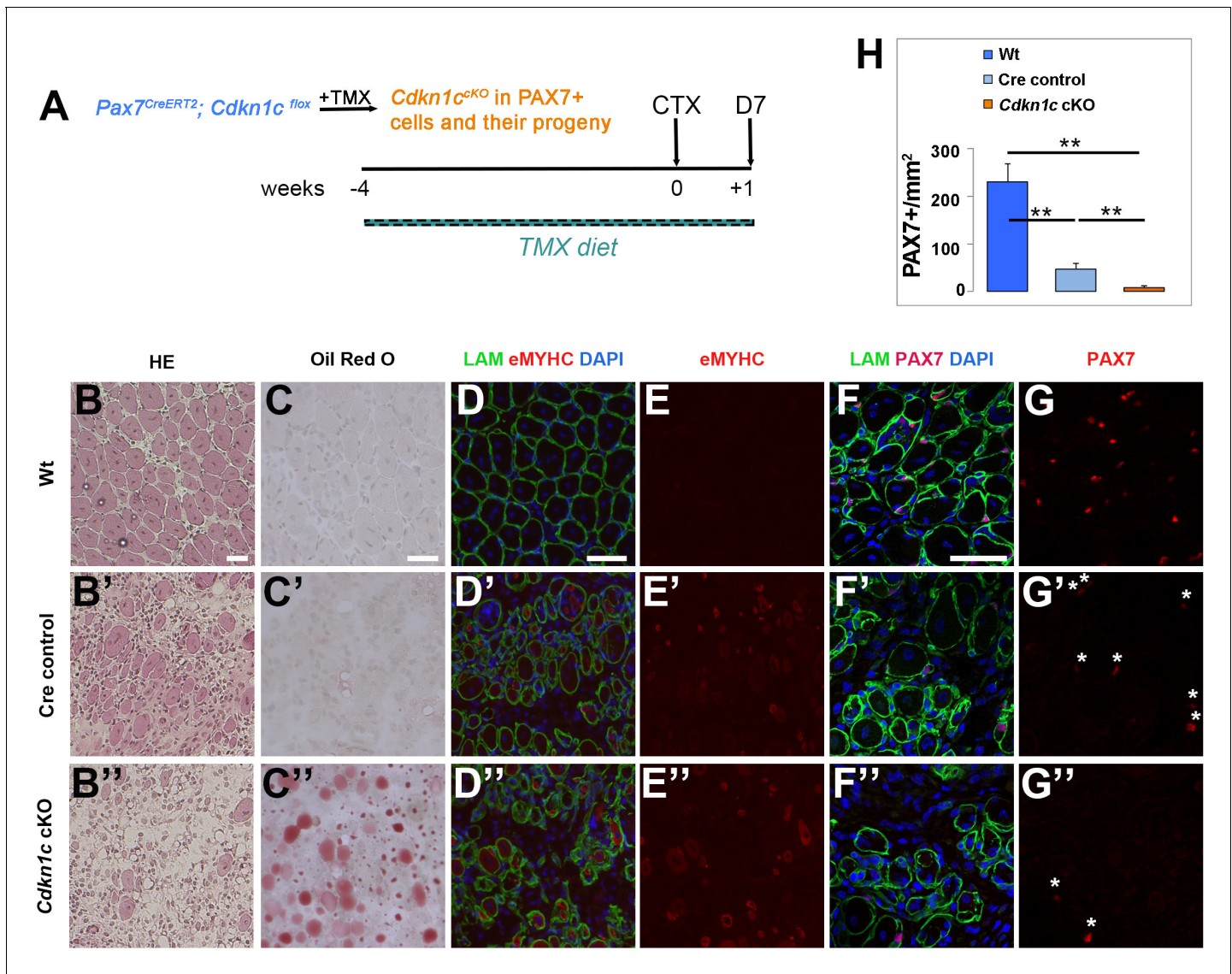

**Figure 4.** MuSC-specific *Cdkn1c* ablation hinders muscle regeneration. (**A**) Time-course of tamoxifen administration, intramuscular injury of TA muscle (CTX arrow), and muscle harvest (D7 arrow). (**B–G**) Cryosections of TA muscle were stained for histological and satellite cell population characterization 7 days after CTX injection. Analyzed animals at (**B-G**) were wild-type littermates (Wt; *Pax7+*; *Cdkn1c+*; **B–G**), Cre control (*Pax7^CreERT2*; **B'–G'**), and *Cdkn1c* cKO (*Pax7^CreERT2*; *Cdkn1c^Flox*; **B''–G''**). (**B**) HE staining for histologic characterization of the muscles. (**C**) Oil Red O staining for evaluation of fat infiltration of the muscles. (**D–E**) embryonic myosin (eMYHC, red)/LAMININ (LAM, green) immunofluorescence to mark newly formed myofibers post-regeneration. (**F–G**) PAX7 (red)/LAMININ (LAM, green) immunofluorescence to mark PAX7+ satellite cells. Nuclei in (**D-G**) were counter-stained with DAPI (blue). Scale bars, 50 µm. (**H**) Quantification of (**F-G**). Data show mean +SD, n ≥ 5 animals. Asterisks indicate significance; **p≤0.01.

DOI: https://doi.org/10.7554/eLife.33337.009

The following figure supplement is available for figure 4:

**Figure supplement 1.** In vivo MuSC-specific Cdkn1c ablation.

DOI: https://doi.org/10.7554/eLife.33337.010

myoblasts at T24-T48 presented with increasing amounts of CDKN1c. Unexpectedly, this negative cell cycle regulator was not expressed in the quiescent population, but in activated MuSCs entering the cell cycle. However, during this early proliferation phase, CDKN1c was restricted to the cytoplasm (*Figures 5A–B* and *6A–B*). Consistent with these findings, CDKN1c was mainly restricted to the cytoplasm at d3 following CTX injection in vivo, a time point when most cells are in a proliferative state (*Figure 5C*).

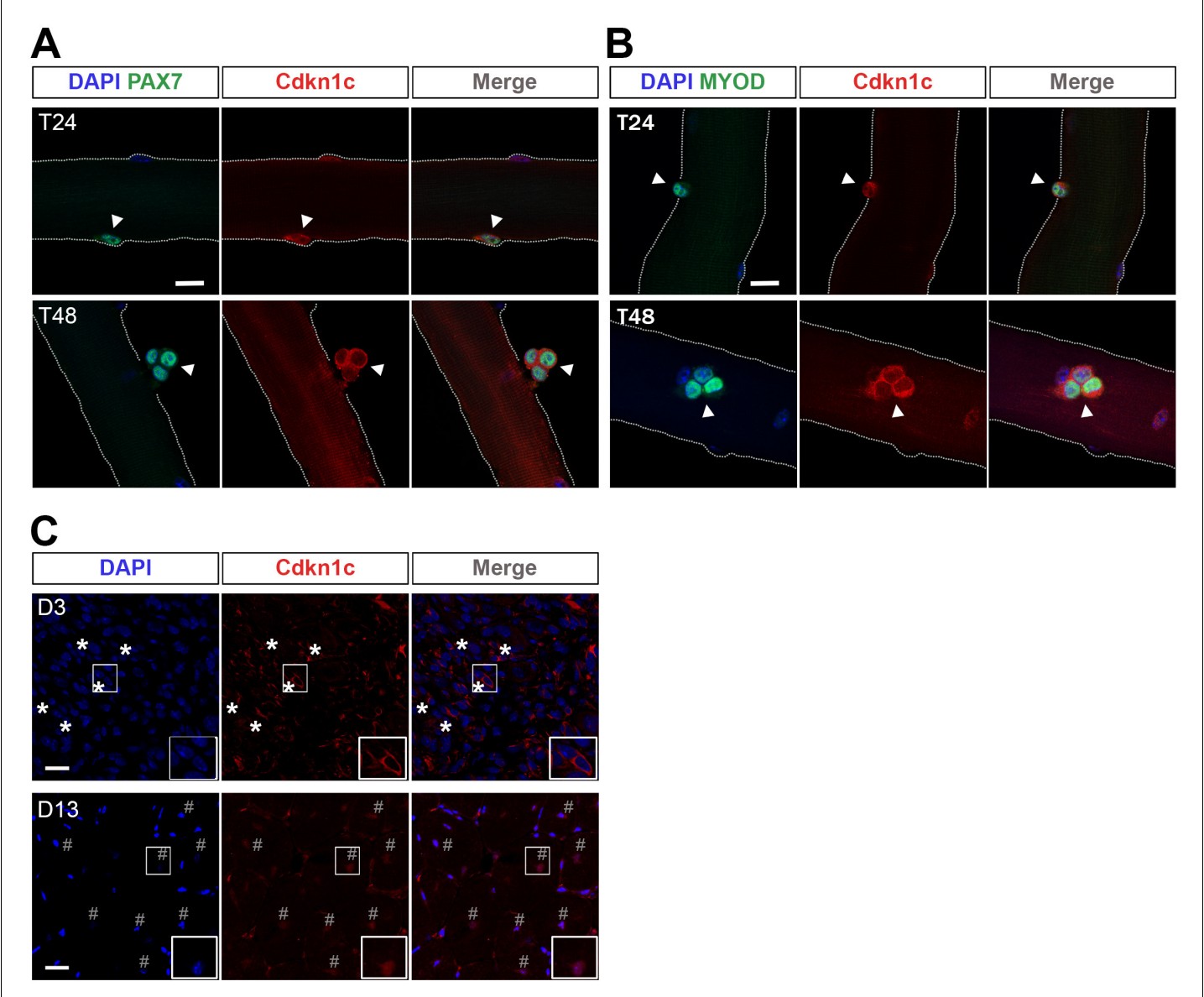

**Figure 5.** Cdkn1c cytoplasmic expression after satellite cell activation. (**A**) Satellite-cell-derived myoblasts (**T24–T48**) of single EDL myofibers stained with PAX7 (green) and CDKN1c (red). Arrowheads indicate PAX7+ cells. (**B**) Satellite cell-derived myoblasts of single EDL myofibers stained with MYOD (green) and CDKN1c (red). Arrowheads indicate MYOD+ cells. Nuclei were counter-stained with DAPI. n ≥ 3. Scale bars, 40 μm. (**C**) CDKN1c (red) staining of TA muscle at 3 (**D3**) or t (**D13**) days after CTX injection. Asterisks indicate regions with cytoplasmic CDKN1c. # indicates central nuclei of newly formed fibers during muscle regeneration. Nuclei were counter-stained with DAPI. Scale bars, 20 μm.
DOI: https://doi.org/10.7554/eLife.33337.011

The following figure supplement is available for figure 5:

**Figure supplement 1.** Cdkn1c is not expressed in quiescent satellite cells.
DOI: https://doi.org/10.7554/eLife.33337.012

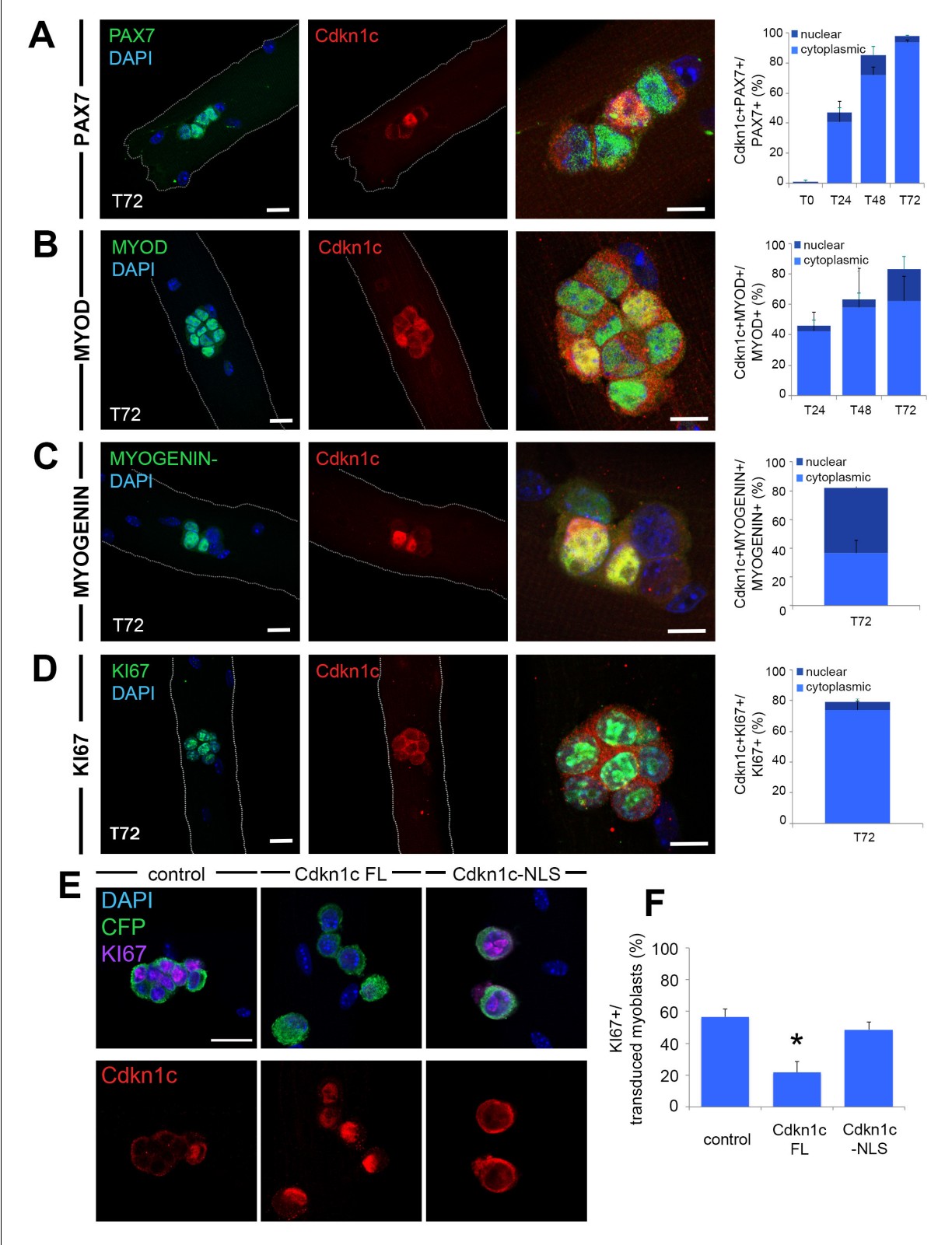

**Figure 6.** Cdkn1c expression and subcellular localization during satellite cell activation and differentiation. (**A–D**) Immunofluorescence for PAX7 (A; green), MYOD (B; green), MYOGENIN (C; green) or KI67 (D; green) and Cdkn1c (red) at T72 in single myofiber cultures of EDL muscles and quantification of PAX7+ (**A**), MYOD+ (**B**), MYOGENIN+ (**C**) or KI67+ (**D**) cells that co-expressed CDKN1c over the time-course of the culture. Cytoplasmic (light blue) or nuclear (dark blue) localization of CDKN1c is indicated in the graphs. Scale bars, 40 μm. (**E**) Immunofluorescence for Cyant

*Figure 6 continued on next page*

*Figure 6 continued*

fluorescent protein (CFP, green), KI67 (purple), and CDKN1c (red) in transduced (i.e. CFP+) myoblasts at T72 in single myofiber cultures. Fibers were transduced with empty retroviruses (control; left panel), retroviruses expressing full-length *Cdkn1c* (Cdkn1c FL; middle panel) or Nuclear localization signal-deficient *Cdkn1c* (Cdkn1c–NLS; right panel). (F) Quantification of transduced (CFP+) myoblasts that were proliferating (KI67+). Nuclei were counter-stained with DAPI (blue). Scale bars, 20 μm. Data show mean +SD, n ≥ 3 animals, 20–32 fibers/animal. *p≤0.05 compared to control virus.
DOI: https://doi.org/10.7554/eLife.33337.013

The following figure supplements are available for figure 6:

**Figure supplement 1.** Lack of CDKN1c binding in myogenic regulatory regions.
DOI: https://doi.org/10.7554/eLife.33337.014

**Figure supplement 2.** Uncoupling of cell cycle exit and differentiation.
DOI: https://doi.org/10.7554/eLife.33337.015

At T72, clusters composed of cells at different states are observed, including differentiation (loss of PAX7 expression associated with expression of MYOD and MYOGENIN) and self-renewal (maintenance of PAX7, loss of MYOD, and lack of MYOGENIN) (*Zammit et al., 2004*). Thus, there is a mixed population of self-renewing (cells expressing PAX7 alone), activated/proliferating (cells co-expressing PAX7 and MYOD), and differentiating (cells expressing MYOGENIN and/or MYOD) myogenic cells. At this stage, we observed high percentages of CDKN1c expression in each of these populations (*Figure 6A–C*). Interestingly, as differentiation proceeded (MYOD+ followed by MYOGENIN +at T72), Cdkn1c expression became increasingly nuclear (*Figure 6B–C*). Although Cdkn1c was mostly cytoplasmic in PAX7+/Cdkn1c + T72 myoblasts, Cdkn1c exhibited nuclear presence in around 25% of MYOD+/Cdkn1c+ and 55% of MYOGENIN+/Cdkn1c + T72 myoblasts (*Figure 6A–C*). In line with this, Cdkn1c protein was mainly restricted to cell nuclei at late stages of regeneration following in vivo muscle injury (d13 post-CTX; *Figure 5C*). Finally, similarly to the T24-T48 activated/cycling populations of myoblasts in single myofibers ex vivo, Cdkn1c continued to be present in KI67+ proliferating cells at T72, yet limited to their cytoplasm (*Figure 6D*). Next, to evaluate whether Cdkn1c nuclear translocation was linked with its association with MYOD (*Reynaud et al., 2000*; *Vaccarello et al., 2006*; *Figliola et al., 2008*; *Osborn et al., 2011*; *Busanello et al., 2012*; *Battistelli et al., 2014*; *Zalc et al., 2014*), we used data generated by a MYOD ChIP sequencing experiment (*Cao et al., 2010*) to identify MYOD-binding sites at muscle regulatory regions. We performed ChIP experiments with an anti-Cdkn1c antibody to test whether Cdkn1c shares these binding sites with MYOD, and we found no significant enrichment at any of the tested sites (*Figure 6—figure supplement 1*).

Finally, to establish whether the subcellular localization of CDKN1c directly affects the cycling status of myoblasts, we generated retroviruses encoding full-length *Cdkn1c* (Cdkn1c FL) or *Cdkn1c* lacking the Nuclear localization signal (Cdkn1c –NLS), using Cyan fluorescence protein (CFP) as a reporter to identify transduced cells (*Figure 6E*). A mock CFP retrovirus was used as control (*Figure 6E–F*). Overexpression of Cdkn1c FL in single myofiber cultures was associated with nuclear CDKN1c protein (*Figure 6E*) and a marked decrease in cycling myoblasts (*Figure 6E,F*). In contrast, forced expression of the NLS-deficient construct restricted CDKN1c to the cytoplasm (*Figure 6E*) and did not induce cell cycle exit as Cdkn1c FL (*Figure 6E,F*). By contrast, overexpression of either Cdkn1c FL or Cdkn1c –NLS did not alter the proportions of cells entering myogenic differentiation, as evidenced by MYOD and/or MYOGENIN expression (*Figure 6—figure supplement 2A–D*). Lastly, in order to eliminate potential interference with endogenous Cdkn1cCDKN1c in these experiments, we repeated them in *Cdkn1c*-deficient myoblasts following FACS isolation (*Figure 6—figure supplement 2E*). Comparably to the ex vivo system of single myofibers (*Figure 6E*, *Figure 6—figure supplement 2A–D*), we observed cell cycle exit, uncoupled from myogenic differentiation, in Cdkn1c FL but not Cdkn1c –NLS (*Figure 6—figure supplement 2E–G*). Together, these results show that nuclear CDKN1c promotes growth arrest, while cytoplasmic localization of the protein - observed at early stages of MuSC activation - is compatible with proliferation. Furthermore, our data suggest uncoupling of cell cycle exit and myogenic differentiation.

# Discussion

Regenerative adult myogenesis is crucial for recovery from injuries, but can be compromised by degenerative or disease states that affect the functional capacity of skeletal muscle stem cells (MuSC). The maintenance of MuSC function largely depends on the entry and maintenance of a non-cycling, reversible quiescent state. The molecular mechanisms that control MuSC cell cycle transitions and adult myogenesis have gained significant interest in recent years, as a way to understand post-trauma tissue restoration and, subsequently, to design efficient innovative therapies when it is defective.

Focusing on cell cycle exit signals and given our recent findings on the role of CDKN1c in cell fate decisions in embryonic/fetal myogenesis (*Zalc et al., 2014*), we hypothesized that CDKN1c may control cell cycle and differentiation of MuSCs in adult muscle. Indeed, *Cdkn1c* deficiency compromised muscle regeneration following in vivo injury (*Figure 2A–D*; *Figure 4*) and hindered primary myoblast differentiation and myotube formation (*Figure 3*; *Figure 3—figure supplement 2*). CDKN1c expression correlates with differentiation in many other cell types (*Westbury et al., 2001*), while in myogenic cultures CDKN1c has also been considered as a differentiation marker (*Reynaud et al., 1999*; *Mounier et al., 2011*). Nevertheless, the consequences of its ablation on adult muscle have not been examined previously, notably because of the perinatal lethality of *Cdkn1c* mutant mice. Maintaining the mice in a mixed CD1;B6 background allowed us to obtain some survivors, while analyzing floxed *Cdkn1c* mice allowed cell-specific studies. In vivo and in vitro analysis of both models demonstrated that lack of *Cdkn1c* compromised myogenic differentiation (*Figures 1–4*, *Figure 3—figure supplement 2*). In the case of *Cdkn1c*-null mice, there was increased MuSC self-renewal at the expense of differentiation (*Figures 1–2*), while conditional deletion of *Cdkn1c* in adult mice diminished the MuSC compartment after injury (*Figure 4*). These seemingly contradictory results might depend on compensatory mechanisms that are activated when *Cdnk1c* deletion is induced prenatally (mutant) versus postnatally (cKO). Moreover, in our cKO mice, compounding effects of inactivating both *Pax7* (due to CreERT2 insertion in the PAX7 locus) and *Cdkn1c* cannot be excluded. However, comparing the *Cdkn1c* cKO mice to the Cre control, showed a much more severe regeneration impairment in the former (*Figure 4*). Together, our data on *Cdkn1c* mutant and cKO mice implicate an important function for CDKN1c in adult myogenesis, similarly to its emerging importance in the quiescence and renewal of several other stem cell types [Matsumoto et a., 2011; *Zacharek et al., 2011*; *Furutachi et al., 2013*).

Examining the myoblast populations expressing CDKN1c (i.e. quiescent vs. activated), we did not detect CDKN1c in quiescent MuSCs on sections or isolated myofibers of resting adult (8–12 weeks) muscle (*Figure 5—figure supplement 1*). On the contrary, CDKN1c was readily detected in interstitial cells (*Figure 5—figure supplement 1*). Our finding is consistent with previous reports of high CDKN1c levels in adult muscle (*Matsuoka et al., 1995*; *Park and Chung, 2001*) and lack of CDKN1c in FACS-isolated postnatal MuSC populations (*Chakkalakal et al., 2014*) or myofiber-associated MuSCs (*Naito et al., 2016*). In contrast, an early study detected CDKN1c in quiescent MuSCs (*Fukada et al., 2007*) which were isolated with a FACS protocol using an antibody previously described by the same group (*Fukada et al., 2004*). However, this antibody immuno-reacts with bone marrow cells (*Fukada et al., 2004*), while CDKN1c has a well-established role and presence in the hematopoietic lineage (*Matsumoto et al., 2011*; *Zou et al., 2011*). Furthermore, CDKN1c immunostaining in *Fukada et al. (2007)* was performed with an antibody against the CDKN1c carboxy-terminus, which might cross-react with the respective domain of CDKN1b (*Matsuoka et al., 1995*; *Galea et al., 2008*; *Pateras et al., 2009*). Combining our observation with previous reports (*Fukada et al., 2004*; *Fukada et al., 2007*; *Chakkalakal et al., 2014*; *Naito et al., 2016*); present study], we conclude that quiescent MuSCs do not express CDKN1c. Instead, it is established that they express CDKN1b, another member of the CDKI family including CDKN1c (*Chakkalakal et al., 2014*); our unpublished data].

Upon MuSC activation and differentiation, we show that CDKN1c is strongly upregulated (*Figures 5–6*). Accordingly, activation of MuSCs during muscle regeneration has been shown to induce CDKN1c, with its levels peaking at d3-4 (*Yan et al., 2003*); our unpublished observations]. The early induction of CDKN1c upon activation might be associated with MYOD expression, a transcription factor with which CDKN1c has been implicated in a positive feedback loop (*Reynaud et al., 2000*; *Osborn et al., 2011*). Specifically, MYOD has been shown to induce *Cdkn1c* both by disrupting a

chromatin loop to release the *Cdkn1c* promoter and by upregulating intermediate factors (*Vaccarello et al., 2006*; *Figliola et al., 2008*; *Busanello et al., 2012*; *Battistelli et al., 2014*). Furthermore, we previously identified a muscle-specific *Cdkn1c* regulatory element that MYOD binds and transactivates (*Zalc et al., 2014*), while co-immunoprecipitation assays revealed direct CDKN1c-MYOD binding (*Reynaud et al., 2000*). Moreover, *MyoD* mutant mice present a similar muscle phenotype as our *Cdkn1c*-deficient mice (*Megeney et al., 1996*), also consistent with *Cdkn1c* being a direct downstream gene of MYOD, and possible association of MYOD and CDKN1c. MYOD is expressed within hours after MuSC activation (*Zammit et al., 2004*; *Zhang et al., 2010*) and sustains the transition from quiescence to cell cycle via the replication-related factor CDC6 (*Zhang et al., 2010*). MYOD induces growth arrest in non-myogenic cell lines (*Crescenzi et al., 1990*; *Sorrentino et al., 1990*) and has a well-established role for entry into the myogenic lineage. While the possibility of CDKN1c associating with MYOD to regulate CDKN1c translocation was an attractive hypothesis, CDKN1c was not found associated with MYOD binding in muscle-specific gene regulatory regions (*Cao et al., 2010*) evaluated by ChIP (*Figure 6—figure supplement 1*). Remarkably, despite robust expression of MYOD, myoblasts continue to proliferate and do not proceed to differentiation for several days (*Tajbakhsh, 2009*). In fact, in dividing myoblasts MYOD activity is inhibited by Id proteins and CDK/Cyclin complexes (*Wei and Paterson, 2001*). Furthermore, additional factors, including MYOGENIN, were suggested to initiate or enhance transcription of some MYOD targets (*Blais et al., 2005*; *Cao et al., 2006*).

We demonstrate robust CDKN1c presence in activated MuSCs, including expression in proliferating myoblasts in isolated myofiber cultures (e.g. KI67+ cells at T72, cycling cells at T24-T48; *Figures 5–6*). These observations might contradict its traditional function as cell cycle exit factor (*Lee et al., 1995*; *Matsuoka et al., 1995*) and its role in growth arrest during embryonic myogenesis (*Zhang et al., 1999*; *Zalc et al., 2014*). Our data, however, suggest that the uncoupling of CDKN1c growth arrest activity in proliferating myoblasts of isolated myofiber cultures is related to its subcellular localization. CDKN1c is cytoplasmically restricted in activated myoblasts but is progressively detected in the nucleus as differentiation takes place (*Figure 6*). Indeed, forced CDKN1c expression in myoblasts of isolated myofiber cultures led to nuclear CDKN1c and growth arrest, while overexpression of Nuclear localization signal-deficient CDKN1c was compatible with myoblast proliferation (*Figure 6*; *Figure 6—figure supplement 2*). On the contrary, CDKN1c cyto-nuclear shuttling was not related to myogenic differentiation, suggesting an uncoupling of cell cycle exit and myogenic differentiation (*Figure 6—figure supplement 2*), in line with our previous observations in embryonic muscles (*Zalc et al., 2014*).

We have not identified the molecular events underlying the CDKN1c nucleo-cytoplasmic shuttling, although some observations might explain this pattern. Firstly, CDKN1c could be implicated in cell cycle progression through CDK/Cyclin assembly, similarly to its Cip/Kip siblings (*Michieli et al., 1994*; *Harper et al., 1995*; *LaBaer et al., 1997*; *Peschiaroli et al., 2002*). In the absence of CDKN1a and CDKN1b, CDKN1c is solely responsible for CDK/Cyclin complex stabilization in mouse embryonic fibroblasts (*Cerqueira et al., 2014*). However, this might be less likely in myoblasts, where Cdkn1c-MYOD binding engages the CDKN1c helix domain (*Reynaud et al., 2000*) that was found indispensable for CDK/Cyclin binding and inhibition (*Hashimoto et al., 1998*; *Reynaud et al., 2000*). Moreover, in agreement with CDKN1c function as a cell cycle inhibitor, we observed reduced differentiation and increased proliferation in its absence (*Figures 1I–J*, *2A–E* and *3E–N*). Secondly, CDKN1c could be involved in nucleo-cytoplasmic distribution of cyclins or CDKs, as previously observed in other cell types. CDKN1c has been shown to interfere with the nuclear translocation of cyclin D1 (*Zou et al., 2011*) and to relocalize a fraction of CDK2 into the cytoplasm (*Figliola and Maione, 2004*). Thirdly, while in the cytoplasm, CDKN1c might participate in MuSC mobilization, one of the earliest manifestations of their activation (*Siegel et al., 2009*). Cytoplasmic CDKN1c was described to regulate cell motility together with LIM-kinase1 (*Vlachos and Joseph, 2009*; *Chow et al., 2011*; *Guo et al., 2015*). Although we did not detect LIM-kinase1 in myogenic cells, we cannot exclude association with other, yet uncharacterized, partners. Future studies are expected to elucidate the roles of CDKN1c in different sub-cellular compartments of myoblasts.

In conclusion, our data indicate that CDKN1c plays essential roles at the initial phase following MuSC activation. We suggest that CDKN1c acts at that early phase, as a) in vivo, *Cdkn1c* mutants display a strong phenotype during the first days after injury and b) *Cdkn1c* mutant myoblasts have compromised functions in cultures that last a few days, thus resembling the first events after MuSC

activation. CDKN1c presence is compatible with activation/proliferation and possibly represents an early activation event. Its loss profoundly affects ex vivo myogenic differentiation and delays in vivo post-injury recovery, which may lead to increased MuSC self-renewal. Therefore, the reduced regeneration capacity of *Cdkn1c*-mutant muscle is not caused by decreased numbers of MuSCs, but results from the reduction of myogenic differentiation, which increases propensity for stem-cell self-renewal. Defective stem cell cycle dynamics and continuous activation/proliferation can lead to DNA damage accumulation, apoptosis, pool exhaustion, and inability to support homeostatic or regenerative demands. A better understanding of cell cycle regulation in MuSCs is imperative to define the molecular events underlying their long-term preservation.

# Materials and methods

## Key resources table

| Reagent type (species) or resource | Designation | Source or reference | Identifiers | Additional information |
|---|---|---|---|---|
| Strain, strain background (*M. musculus*) | *Cdkn1c^tm1Sje* | The Jackson Laboratory; PMID: 9144284 | MGI: J40203, RRID:IMSR_JAX:003336 | |
| Strain, strain background (*M. musculus*) | *p57^flox* | PMID: 28196404 | | Mouse line generated by the group of F.Relaix and characterized in *Mademtzoglou et al. (2017)*; Genesis 55(4) doi: 10.1002/dvg.23025 |
| Strain, strain background (*M. musculus*) | *Pax7^CreERT2/+* | The Jackson Laboratory; PMID: 19554048 | MGI: J:150962; RRID:IMSR_JAX:012476 | Mouse line obtained from C.M. Fan |
| Strain, strain background (*M. musculus*) | *Rosa^mTmG* | The Jackson Laboratory; PMID: 17868096 | MGI: J:124702; RRID:IMSR_JAX:007576 | |
| Genetic reagent (synthetic) | *pGEMT-Easy* vector | Promega | A1360 | |
| Cell line (*Homo sapiens*) | 293T | DSMZ | ACC635; RRID:CVCL_0063 | https://www.dsmz.de/catalogues/details/culture/ACC-635.html?tx_dsmzresources_pi5%5BreturnPid%5D=192 |
| Cell line (*M. musculus*) | C2C12 | American Type Culture Collection (ATCC); PMID: 28966089 | CRL-1772; RRID: CVCL_0188 | Cell line maintained in E. Gomes lab |
| Antibody | anti-CD31-PE (monoclonal) | eBiosciences | 12-0311-81; RRID:AB_465631 | |
| Antibody | anti-CD45-PE (monoclonal) | eBiosciences | 12-0451-81; RRID:AB_465667 | |
| Antibody | anti-embryonic MyHC (mouse monoclonal) | DSHB | F1.652; RRID:AB_528358 | |
| Antibody | anti-embryonic MyHC (mouse monoclonal) | Santa Cruz | sc53091; RRID:AB_670121 | |
| Antibody | anti-GFP (chicken polyclonal) | Abcam | ab13970; RRID:AB_300798 | |
| Antibody | anti-integrin a-biotin (mouse) | Miltenyi Biotec | 130-101-979; RRID:AB_2652472 | |
| Antibody | anti-IgG (rabbit) | Diagenode | C15410206 | |
| Antibody | anti-KI67 (mouse monoclonal) | BD Pharmingen | 556003; RRID:AB_396287 | |
| Antibody | anti-Laminin (rabbit polyclonal) | Sigma-Aldrich | L9393; RRID:AB_477163 | |
| Antibody | anti-Laminin (rat monoclonal) | Sigma-Aldrich | 4H8-2; RRID:AB_784266 | |
| Antibody | anti-Laminin (rabbit polyclonal) | Novus Biological | NB300-144AF647 | |

*Continued on next page*

Continued

| Reagent type (species) or resource | Designation | Source or reference | Identifiers | Additional information |
|---|---|---|---|---|
| Antibody | anti-MyHC (mouse monoclonal) | DSHB | mf20-c; RRID:AB_2147781 | |
| Antibody | anti-MyoD (mouse monoclonal) | DAKO | M3512; RRID:AB_2148874 | |
| Antibody | anti-MyoD (rabbit polyclonal) | Santa Cruz | sc-760; RRID:AB_2148870 | |
| Antibody | anti-Myogenin (mouse monoclonal) | DSHB | F5D; RRID:AB_2146602 | |
| Antibody | anti-p57 (goat polyclonal) | Santa Cruz | sc1039; RRID:AB_2078158 | |
| Antibody | anti-p57 (mouse monoclonal) | Santa Cruz | sc56431; RRID:AB_2298043 | |
| Antibody | anti-p57 (rabbit polyclonal) | Santa Cruz | sc8298; RRID:AB_2078155 | |
| Antibody | anti-Pax7 (mouse monoclonal) | DSHB | PAX7-c; RRID:AB_528428 | |
| Antibody | anti-Sca-1-PE (mouse) | eBiosciences | 12-5981-81; RRID:AB_466085 | |
| Antibody | fab fragment affinity-purified antibody (goat) | Jackson ImmunoResearch | 115-007-003 | |
| Sequence-based reagent | AGGGCATATCC AACAACAAACTT | Eurogentec | N/A | qPCR HPRT (Forward primer) |
| Sequence-based reagent | GTTAAGCAGTA CAGCCCCAAA | Eurogentec | N/A | qPCR HPRT (Reverse primer) |
| Sequence-based reagent | CTGAAGGACCA GCCTCTCTC | Eurogentec | N/A | qPCR p57 (Forward primer) |
| Sequence-based reagent | AAGAAGTCGTT CGCATTGGC | Eurogentec | N/A | qPCR p57 (Reverse primer) |
| Sequence-based reagent | ATCTGAGGTCA GCCATTTGGT | Eurogentec | N/A | ChIP qPCR Mef2a (Forward primer) |
| Sequence-based reagent | GCTAAGGACAG CTGTGACCTG | Eurogentec | N/A | ChIP qPCR Mef2a (Reverse primer) |
| Sequence-based reagent | TTAAAGACATGTG GCAACAGACTAC | Eurogentec | N/A | ChIP qPCR Lmn2b (Forward primer) |
| Sequence-based reagent | TGCTCTTTCTGTA CTGTGTGGTG | Eurogentec | N/A | ChIP qPCR Lmn2b (Reverse primer) |
| Sequence-based reagent | GGAGTGATTGA GGTGGACAGA | Eurogentec | N/A | ChIP qPCR Lincmd1 (Forward primer) |
| Sequence-based reagent | CTCTCCCACCTG TTTGTGTCTT | Eurogentec | N/A | ChIP qPCR Lincmd1 (Reverse primer) |
| Sequence-based reagent | AATTACAGCCG ACGGCCTCC | Eurogentec | N/A | ChIP qPCR Myogenin (Forward primer) |
| Sequence-based reagent | CCAACGCCACA GAAACCTGA | Eurogentec | N/A | ChIP qPCR Myogenin (Reverse primer) |
| Sequence-based reagent | CAGCTCCTTG CCCTGTGAAA | Eurogentec | N/A | ChIP qPCR Desmin-proximal (Forward primer) |
| Sequence-based reagent | TGTAGCCCTCC TGACATCAC | Eurogentec | N/A | ChIP qPCR Desmin proximal (Reverse primer) |
| Sequence-based reagent | CCAAAAGGG CCGATGAGGAA | Eurogentec | N/A | ChIP qPCR Desmin distal (Forward primer) |
| Sequence-based reagent | TAGAGACAGA CCAGTGGCGG | Eurogentec | N/A | ChIP qPCR Desmin distal (Reverse primer) |

Continued

| Reagent type (species) or resource | Designation | Source or reference | Identifiers | Additional information |
|---|---|---|---|---|
| Commercial assay or kit | LightCycler 480 SYBR Green I Master | Roche-Sigma-Aldrich | 04887352001 | |
| Commercial assay or kit | iDeal ChIP-seq kit | Diagenode | C01010051 | |
| Commercial assay or kit | RNasy Micro Kit | QIAGEN | 74004 | |
| Commercial assay or kit | Transcriptor First Strand cDNA Synthesis Kit | Roche-Sigma-Aldrich | 4379012001 | |
| Chemical compound, drug | bFGF | Peprotech | 450–33 | 20 ng/ml |
| Chemical compound, drug | bFGF | Thermo Fisher Scientific | PHG0263 | 20 ng/ml |
| Chemical compound, drug | Bovine serum albumin (BSA) | Jackson ImmunoResearch | 10001620 | 0.2% |
| Chemical compound, drug | Cardiotoxin | Latoxan | L8102 | 10 µM |
| Chemical compound, drug | Cardiotoxin | Sigma-Aldrich | 217503–1 mg | 10 µM |
| Chemical compound, drug | Chicken embryo extract | MP-Biomedical | 92850145 | 0.5% |
| Chemical compound, drug | Chicken embryo extract | Seralab | CE-650-J | 1% |
| Chemical compound, drug | collagen | BD Biosciences | 354236 | culture dish coating |
| Chemical compound, drug | Collagenase type I | Sigma-Aldrich | C0130 | 0.2% |
| Chemical compound, drug | Collagenase type I | Worthington Biochemical Corp | 9001-12-1 | |
| Chemical compound, drug | Collagenase A | Roche-Sigma-Aldrich | 11088793001 | 0.2% w/v |
| Chemical compound, drug | DAPI (4′,6-diamidino-2-phenylindole dihydrochloride) | Thermo Fisher Scientific | D1306 | |
| Chemical compound, drug | Dispase II | Roche-Sigma-Aldrich | 4942078001 | 2.4 U/ml |
| Chemical compound, drug | DNaseI | Roche-Sigma-Aldrich | 11284932001 | 10 ng/mL |
| Chemical compound, drug | Dulbecco's Modified Eagle's Medium (DMEM) | Thermo Fisher Scientific | 41966 | single myofiber culture |
| Chemical compound, drug | DMEM with GlutaMAX | Thermo Fisher Scientific | 61965 | myoblast culture |
| Chemical compound, drug | EdU | Thermo Fisher Scientific | C10340 | 2 µM |
| Chemical compound, drug | F-10 Ham's media | Sigma-Aldrich | N6635 | N/A |
| Chemical compound, drug | Fetal bovine serum (FBS) | Thermo Fisher Scientific | 10270 | 20% |
| Chemical compound, drug | Fetal calf serum (FCS) | Eurobio | CVFSVF00-01 | 10% (prol/tion medium), 2% (diff/tion medium) |
| Chemical compound, drug | Fluoromount-G | Southern Biotech | 0100–01 | |
| Chemical compound, drug | Hanks' Balanced Salt Solution (HBSS) | Thermo Fisher Scientific | 14025 | |
| Chemical compound, drug | Hepes | Thermo Fisher Scientific | 15630 | 0.1M |
| Chemical compound, drug | Horse serum | Thermo Fisher Scientific | 26050088 | 5% (coating), 10% (culture) |
| Chemical compound, drug | L-glutamine | Thermo Fisher Scientific | 25030 | 20 mM |
| Chemical compound, drug | matrigel | Corning Life Sciences | 354230 | 1:20 in DMEM |
| Chemical compound, drug | Penicillin/streptomycin | Life Technologies | 15140 | 1X |
| Chemical compound, drug | Pyruvate | Thermo Fisher Scientific | 11360 | 10 mM |
| Software, algorithm | Photoshop CS5 | https://www.adobe.com/products/photoshop.html | RRID:SCR_014199 | |
| Other (anti-biotin beads) | anti-biotin beads | Miltenyi Biotec | 130-090-485; RRID:AB_244365 | MACS |
| Other (anti-PE beads) | anti-PE beads | Miltenyi Biotec | 130-048-801; RRID:AB_244373 | MACS |
| Other (chamber slides) | chamber slide | Nalge Nunc International | 177445 | myoblast culture |

Continued

| Reagent type (species) or resource | Designation | Source or reference | Identifiers | Additional information |
|---|---|---|---|---|
| Other (culture plates) | petri dish | Sigma-Aldrich | Z692301 | single myofiber culture |
| Other (LD column) | LD column | Miltenyi Biotec | 130-042-901 | MACS |
| Other (MS column) | MS column | Miltenyi Biotec | 130-042-201 | MACS |
| Other (grip strength meter) | grip strength meter | Columbus Instruments | 1027CSM-D54 | |

## Mouse lines

The following mouse lines have been previously described: $Pax7^{CreERT2/+}$ (*Lepper et al., 2009*), $Rosa^{mTmG}$ (The Jackson Laboratory, stock 007576), $Cdkn1c^{cKO\ (m)/+}$ ($Cdkn1c$ is imprinted with preferential expression of the maternal allele; superscript (m) indicates maternal inheritance) (*Mademtzoglou et al., 2017*), $Cdkn1c^{+/-}$ (*Zhang et al., 1997*). $Cdkn1c^{+/-}$female mice (C57BL/B6J background, Jackson Laboratory) were bred with CD1 (Evigo) male mice to generate $Cdkn1c^{m/-}$ CD1/B6 hybrid mice. Non-mutant littermates were used as controls. For recombination induction with the $Pax7^{CreERT2}$ allele, mice were fed in tamoxifen diet (TD.55125.I, Envigo). C57BL/6J (Janvier) mice were used as wild-type animals for *Figures 5–6*. Adult mice of 8 to 16 weeks of age were used. At least three mice per genotype were assessed. All animals were maintained inside a barrier facility, and all in vivo experiments were performed in accordance with the French and European Community guidelines (File No: 15–018 from the Ethical Committee of Anses/ENVA/UPEC) and Institutional Animal Care and the Use Committee of University of Minnesota (1604-33660A) for the care and use of laboratory animals.

## Single myofiber isolation and culture

Single muscle fibers were isolated by enzymatic digestion and mechanical disruption of *EDL* muscles (*Moyle and Zammit, 2014*). For enzymatic digestion, muscles were incubated for 90 min at 37°C with 0.2% collagenase type I (C0130, Sigma-Aldrich) in Penicillin/Streptomycin(P/S)-supplemented DMEM (41966, ThermoFisher Scientific). For mechanical disruption, muscles were transferred to 5%-horse-serum-coated deep petri dishes (Z692301, Sigma-Aldrich) with P/S-supplemented DMEM and medium was flushed against the muscle. Detached fibers were transferred into new dishes with P/S-supplemented DMEM. For timed culture, all fibers were transferred after finishing isolation into 5%-horse-serum-coated six-well plates and cultured in the presence of 10% horse serum and 0.5% chicken embryo extract (#092850145, MP Biomedicals). Single myofibers for *Figure 6E–F* and *Figure 6—figure supplement 2* were cultured in the presence of 20% FBS (#10270, Life Technologies) and 1% chicken embryo extract (#CE-650-J, Seralab).

## Retroviral cloning and infection

Cdkn1c cDNA was cloned by PCR from mouse cDNA and subcloned in the *pGEMT-Easy* vector (#A1360, Promega). An additional primer was also designed to produce Cdkn1c cDNA lacking the NLS and following sequences (*Lee et al., 1995*). Full-length or -NLS Cdkn1c were subsequently subcloned -using XhoI and EcoRI added to cloning primers- into the MISSINCK retroviral production vector, which was based on the *pMSCV- IRES-eGFP* (MIG) vector (*Pear et al., 1998*), substituting eGFP with an insulin signal sequence-Cyan Fluorescent Protein (CFP)-KDEL sequence in order to restrict fluorescent tracker expression to the endoplasmic reticulum and Golgi (*Alonso-Martin et al., 2016*). MISSINCK retroviruses expressing either full-length or -NLS Cdkn1c were produced in 293T (DSMZ, #ACC635) cells, and retroviral supernatant was acquired 50 hr post-transfection of 293 T cells. For transduction, single myofiber cultures were incubated with 1:10 diluted retroviral supernatant from 24 to 72 hr after fiber isolation (*Figure 6E*; *Figure 6—figure supplement 2A–D*) and primary myoblasts were incubated with 1:10 diluted retroviral supernatant from 48 to 120 hr after FACS isolation (*Figure 6—figure supplement 2E–G*).

## Cell sorting and culture

Using the tamoxifen-inducible Cre line $Pax7^{CreERT2}$, membrane-GFP is expressed in muscle satellite cells (MuSCs) of $Pax7^{CreERT2}$; $Rosa^{mTmG}$ mice. Hindlimb muscles were dissociated by 0.2% w/v

collagenase A (11088793001, Roche) and 2.4 U/ml dispase II (04942078001, Roche) in digestion buffer [HBSS (14025, Thermo Scientific), 1% Penicillin/Streptomycin, 10 ng/ml DNase I (11284932001, Sigma), 0.4 mM $CaCl_2$, 5 mM $MgCl_2$, 0.2% bovine serum albumin (BSA; 0010001620, Jackson ImmunoResearch)] with 90 min incubation at 37°C. Dissociated muscles were filtered through 100 µm and 40 µm cell strainers. GFP +cells were collected based on gating of GFP signal.

Sorted cells were plated on matrigel (354230, Corning Life Sciences)-coated chamber slides (177445, Nalge Nunc International). They were initially cultured in high-serum conditions (referred to as 'growth phase') with 20% fetal bovine serum, 10% horse serum and 1:4000 bFGF (20 ng/ml; 450-33B, PeproTech) in DMEM + Glutamax (61965, ThermoFisher Scientific) supplemented with 1% P/S, 20 mM L-glutamine (25030, Thermo Scientific), 10 mM pyruvate (11360, Thermo Scientific), and 0.1M Hepes (15630, Thermo Scientific). Upon reaching 70% confluence at 5 days post-plating, they were switched to low-serum conditions (5% horse serum in P/S-supplemented DMEM + Glutamax) to differentiate (referred to as 'differentiation phase'). EdU (2 µM; C10340, Thermo Fisher Scientific) chase was performed for 2 hr. EdU-incorporating cells were detected according to the manufacturer's protocol.

## Myoblast culture

Hindlimb muscle of control or *Cdkn1c* mutant mice was harvested for isolation of MuSC-derived primary myoblasts as described previously (*Asakura et al., 2001*; *Motohashi et al., 2014*). Briefly, muscle was dissected, minced and then digested in type I collagenase (Worthington Biochemical Corp.). The dissociated cells were filtered, and incubated with *anti*-CD45-PE, *anti*-Sca-1-PE, and *anti*-CD31-PE antibodies (all from eBiosciences) and integrin α-biotin antibody (Miltenyi Biotec) followed by anti-PE beads (Miltenyi Biotec). For magnetic-activated cell sorting (MACS), LD column (Miltenyi Biotec) was used for negative selection to remove non-muscle cells. Flow-through fraction was used for incubation with anti-biotin beads (Miltenyi Biotec) followed by MS column (Miltenyi Biotec). Enriched MuSCs were fractionated as a positive fraction. MuSC-derived primary myoblasts were cultured in growth media, F-10 Ham's media supplemented with 20% FBS, 20 ng/ml basic fibroblast growth factor (Invitrogen) and 1% penicillin-streptomycin (Gibco) on culture dishes coated with collagen (BD Biosciences), and the growth medium was changed every two days. Procedures for EdU experiments and myogenic differentiation were described above.

## Gene expression analysis

RNA was extracted with the RNeasy Micro kit (74004, Qiagen), according to the manufacturer's instructions. cDNA was synthetized with the Transcriptor First Strand cDNA Synthesis kit (04379012001, Roche). RT-qPCR reactions were carried out in triplicate using the LightCycler 480 SyBR Green I Master (04887352001, Roche). Hypoxanthine Phosphoribosyltransferase 1 (HPRT) transcripts were used for normalization. Oligonucleotides sequences were AGGGCATATCCAACAA-CAAACTT and GTTAAGCAGTACAGCCCCAAA for *HPRT* and CTGAAGGACCAGCCTCTCTC and AAGAAGTCGTTCGCATTGGC for *Cdkn1c*.

## C2C12 culture and ChIP-qPCR

C2C12 (PMID: 28966089) cells were grown in DMEM High Glucose (#41966, Life Technologies) supplemented with 10% SVF (Eurobio) until reaching 70% confluence and then they were changed to low serum (2% SVF), differentiation, conditions for four days. Cells were collected and processed for ChIP with the iDeal ChIP-seq kit (C01010051, Diagenode) according to manufacturer's instructions. 1 µg of a mouse anti-Cdkn1c (sc56341, Santa Cruz) or IgG negative control (C15410206, Diagenode) was used. The precipitated and input chromatins were analyzed by qPCR. Primer sequences are listed in *Table 1*.

## Muscle regeneration

Adult (12- to 15-week-old) mice in *Figure 2* and *Figure 2—figure supplement 1* were intramuscularly injected with 70 µl of 10 µM cardiotoxin (CTX) solution (Sigma-Aldrich) into the *Tibialis Anterior* (TA) after being anesthetized by i.p. injection of Avertin (Sigma-Aldrich) (*Asakura et al., 2002*). Muscles were recovered 3, 4, 7 or 30 days post-injury. Five mg/kg of EdU was injected at 24 hr before sacrifice for EdU staining.

**Table 1.** Sequences of primers used for the ChIP-qPCR.

| Regulated gene/region | Forward primer | Reverse primer |
| --- | --- | --- |
| Mef2a | ATCTGAGGTCAGCCATTTGGT | GCTAAGGACAGCTGTGACCTG |
| Lmn2b | TTAAAGACATGTGGCAACAGACTAC | TGCTCTTTCTGTACTGTGTGGTG |
| Lincmd1 | GGAGTGATTGAGGTGGACAGA | CTCTCCCACCTGTTTGTGTCTT |
| Myogenin | AATTACAGCCGACGGCCTCC | CCAACGCCACAGAAACCTGA |
| Desmin (proximal) | CAGCTCCTTGCCCTGTGAAA | TGTAGCCCTCCTGACATCAC |
| Desmin (distal) | CCAAAAGGGCCGATGAGGAA | TAGAGACAGACCAGTGGCGG |

DOI: https://doi.org/10.7554/eLife.33337.016

Adult (8- to 12-week old) mice in *Figure 4* and *Figure 5C* were intramuscularly injected with 45 ul of 10 μM CTX (Latoxan) into the TA after being anesthetized by i.p. injection of ketamine-xylasin. Muscles were recovered at 3, 7 or 13 days post-injury.

## Grip strength test

The maximum grip strength using a *grip strength* meter (Columbus Instruments) was determined by taking the average of the three highest values out of the 15 values collected and normalized by body weight (*Aartsma-Rus and van Putten, 2014*). We performed three sets of five consecutive measurements for one set, and mice were allowed to rest for at least 20 min between the sets.

## Immunohistochemistry

Sections: Muscles were frozen fresh in liquid nitrogen-cooled isopentane and sectioned at 8 μm. Frozen sections were fixed with 4% paraformaldehyde/PBS for 20 min at room temperature. For hematoxylin-eosin staining, nuclei were stained with hematoxylin (Sigma-Aldrich) for 11 min and cytoplasmes were counter-stained with eosin (Sigma-Aldrich) for 30 s. The sections were then dehydrated with brief passages through increasing concentrations of ethanol (30%, 50%, 70%, 85%, 95%, 100%). Sirius red staining (Sigma-Aldrich) was performed for detection of fibrosis as described previously (*Shimizu-Motohashi et al., 2015*). For two-color immunofluorescence, frozen sections were permeabilized with 0.2% Triton X (Sigma-Aldrich) and blocked with M.O.M kit (Vector laboratories) followed by 2% bovine serum albumin (Sigma-Aldrich) at room temperature. For three-color immunofluorescence, sections were permeabilized and blocked with 3% BSA, 10% lamb serum, 0.25% TritonX-100/PBS for 30 min at room temperature. In both cases, immunolabeling was performed at 4°C overnight for primary antibodies and at room temperature for 1 hr for secondary antibodies. To outline fibers with Alexa-conjugated anti-laminin, incubation was performed for 3 hr at room temperature, after washing out the secondary antibody. Nuclei were counterstained blue with DAPI. When mouse-raised antibodies were applied, endogenous mouse IgG was blocked by incubation with goat anti-mouse fab fragment affinity-purified antibody (115-007-003, Jackson Immunoresearch) for 30 min at room temperature.

Single myofibers: After isolation (T0) or following culture (T24, T48, T72), myofibers were fixed with 37°C-preheated 4% paraformaldehyde/PBS for 10 min at room temperature. Fixed fibers were permeabilized with 0.5% TritonX-100/PBS for 8 min, blocked with 10% goat serum, 10% swine serum in 0.025% Tween20/PBS for 45 min and incubated with primary antibody (overnight at 4°C) and secondary antibody (1 hr at room temperature). Nuclei were counterstained blue with DAPI.

Primary myoblast culture: Cell cultures were fixed with 4% paraformaldehyde/PBS for 15 min at room temperature, permeabilized with 0.5% TritonX-100/PBS for 5 min, blocked with 5% BSA, 10% goat serum and immunolabeled with primary antibody (overnight at 4°C) and secondary antibody (1 hr at room temperature). Nuclei were counterstained blue with DAPI.

## Antibodies

The following antibodies were used: chicken anti-GFP 1:1000 (#ab13970, Abcam), mouse anti-KI67 1:80 (#556003, BD Pharmingen), mouse anti-MYOD 1:80 (M3512, DAKO), rabbit anti-MYOD 1:1000 (sc304, Santa Cruz), mouse anti-MYOGENIN 1:100 (F5D-c, DSHB), mouse anti-embryonic MyHC 1:50 (F1.652, DSHB), mouse anti-embryonic MyHC 1:300 (sc53091, Santa Cruz), mouse anti-MyHC 1:100

(mf20-c, DSHB), rabbit AlexaFluor647-conjugated anti-laminin 1:200 (NB300-144AF647, Novus Biological), rabbit anti-laminin 1:400 (L9393, Sigma Aldrich), rat anti-laminin 1:1000 (4H8-2, Sigma-Aldrich), mouse anti-PAX7 1:100 (Pax7-c, DSHB), rabbit anti-CDKN1c 1:100 (sc8298, Santa Cruz), goat anti-CDKN1c 1:50 (sc1039, Santa Cruz), AlexaFluor-coupled secondary antibodies (Life Technologies, Jackson ImmunoResearch).

## Graphic editing

Graphs and representative photos were arranged in Figure format with the graphics editor Photoshop CS5. Curves were adjusted in some photos with identical adjustments between control and experimental samples. Color intensities of hematoxylin-eosin photos were adjusted to acquire uniform result among different sections.

## Statistical test

Data of control and mutant mice were compared with the Mann-Whitney U-test, using a significance level of 0.05.

## Acknowledgements

We thank Frédéric Auradé for scientific discussions. FR Laboratory is supported by funding from: Association Française contre les Myopathies (AFM) via TRANSLAMUSCLE (PROJECT 19507), Labex REVIVE (ANR-10-LABX-73), Fondation pour la Recherche Médicale (FRM; Grant DEQ20130326526), MYOGRAD network (GK1631), Agence Nationale pour la Recherche (ANR) grant Bone-muscle-repair (ANR-13-BSV1-0011-02), BMP-Myomass (ANR-12-BSV1-0038- 04), Satnet (ANR-15-CE13-0011-01), and RHU CARMMA (ANR-15-RHUS-0003). AA Laboratory is supported by funding from the NIH R01 (1R01AR062142) and NIH R21 (1R21AR070319). No funding sources were involved in study design, data collection and interpretation, or the decision to submit the work for publication. We wish to acknowledge Adeline Henry and Aurélie Guguin (Plateforme de Cytométrie en flux, Institut Mondor de Recherche Biomédicale), Bénédicte Hoareau (Flow Cytometry Core CyPS, Pierre and Marie Curie University), Serban Morosan, and the animal care facility (Centre d'Expérimentation Fonctionnelle, School of Medicine Pierre et Marie Curie).

## Additional information

### Funding

| Funder | Grant reference number | Author |
|---|---|---|
| Association Française contre les Myopathies | TRANSLAMUSCLE (PROJECT 19507) | Frederic Relaix |
| Labex Revive | ANR-10-LABX-73 | Frederic Relaix |
| Fondation pour la Recherche Médicale | Grant DEQ20130326526 | Frederic Relaix |
| MyoGrad Network | GK1631 | Frederic Relaix |
| Agence Nationale de la Recherche | Bone-muscle-repair (ANR-13-BSV1- 455 0011-02),BMP-Myomass (ANR-12-BSV1-0038- 04),Satnet (ANR-15-CE | Frederic Relaix |
| National Institutes of Health | NIH R01 (1R01AR062142) and 457 NIH R21 (1R21AR070319) | Atsushi Asakura |

The funders had no role in study design, data collection and interpretation, or the decision to submit the work for publication.

## Author contributions
Despoina Mademtzoglou, Formal analysis, Investigation, Visualization, Methodology, Writing—original draft, Writing—review and editing; Yoko Asakura, Formal analysis, Investigation, Methodology; Matthew J Borok, Formal analysis, Investigation, Methodology, Writing—review and editing; Sonia Alonso-Martin, Philippos Mourikis, Methodology, Writing—review and editing; Yusaku Kodaka, Formal analysis, Investigation, Visualization, Methodology; Amrudha Mohan, Methodology; Atsushi Asakura, Conceptualization, Resources, Supervision, Funding acquisition, Investigation, Visualization, Writing—review and editing; Frederic Relaix, Conceptualization, Resources, Supervision, Funding acquisition, Investigation, Writing—original draft, Writing—review and editing

## Author ORCIDs
Despoina Mademtzoglou (iD) http://orcid.org/0000-0002-4494-7234
Yoko Asakura (iD) http://orcid.org/0000-0003-4107-4236
Matthew J Borok (iD) http://orcid.org/0000-0003-2951-6265
Sonia Alonso-Martin (iD) http://orcid.org/0000-0002-3254-0365
Philippos Mourikis (iD) http://orcid.org/0000-0003-3164-7109
Yusaku Kodaka (iD) http://orcid.org/0000-0001-7993-9640
Amrudha Mohan (iD) http://orcid.org/0000-0002-7368-9350
Atsushi Asakura (iD) http://orcid.org/0000-0001-8078-1027
Frederic Relaix (iD) http://orcid.org/0000-0003-1270-1472

## Ethics
Animal experimentation: All animals were maintained inside a barrier facility, and all in vivo experiments were performed in accordance with the French and European Community guidelines (File No: 15-018 from the Ethical Committee of Anses/ENVA/UPEC) and Institutional Animal Care and the Use Committee of University of Minnesota (1604-33660A) for the care and use of laboratory animals

## Decision letter and Author response
Decision letter https://doi.org/10.7554/eLife.33337.020
Author response https://doi.org/10.7554/eLife.33337.021

# Additional files
## Supplementary files
• Transparent reporting form
DOI: https://doi.org/10.7554/eLife.33337.017

## Data availability
All data generated or analysed during this study are included in the manuscript and supporting files.

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
