## [Decision Letter]

Thank you for submitting your article "Cellular localization of the cell cycle inhibitor p57kip2 controls proliferation and growth arrest of adult skeletal muscle stem cells" for consideration by *eLife*. Your article has been reviewed by three peer reviewers, one of whom is a member of our Board of Reviewing Editors, and the evaluation has been overseen by Didier Stainier as the Senior Editor. The reviewers have opted to remain anonymous.

The reviewers have discussed the reviews with one another and the Reviewing Editor has drafted this letter to draw your attention to significant concerns that you would need to address before this work could be considered for publication.

Summary:

This manuscript describes a role for the cell-cycle inhibitor p57 in adult skeletal muscle stem cells (MuSCs). Previous work by the same authors indicated that p57 deletion during embryonic development leads to defects in muscle progenitors' proliferation and differentiation. Similarly, a role for p57 in the proliferation and differentiation of the myogenic C2C12 cell line has been reported.

Here, the authors report that a very small percentage (4.2%) of germline p57 mice survived to adulthood in a mixed background. The surviving animals displayed reduced body weight, decreased myofiber cross-sectional area and increased fibrosis. A marginal increase (1%) of Pax7+ MuSCs was observed on freshly isolated fibers of p57 knock-out mice. Such small increase in Pax7+ MuSCs persisted 30 days after muscle injury. At this time point, no statistically significant difference was noted in Pax7+/MyoD+ committed muscle progenitors of p57 knock-out versus control. To circumvent the lethality observed in germline p57 knock-out mice, the authors selectively deleted p57 in MuSCs by crossing maternal p57fl(m)/+ with tamoxifen-inducible Pax7-CreERT2 mice. Unfortunately, p57 was not efficiently ablated at high frequency. FACS-isolated p57-deleted MuSCs had increased proliferation and decreased differentiation potentials. Quiescent MuSCS did not express p57 which was detected in activated in MuSCs.

Subcellular localization of p57 progressively shifted from the cytoplasm during activation to the nucleus upon differentiation. Overexpression of full-length p57, but not of a mutant incapable of nuclear localization, negatively regulates cell proliferation.

Essential revisions:

1) Beside causing perinatal lethality, germline p57 ablation induces gross developmental defects owing to its role in controlling quiescence of hematopoietic and neural stem cells. Thus, the combination of very few (~4%) p57 knock-out surviving animals and the extensive non-cell specific function of p57 make the results of the presented experiments difficult to interpret.

2) To circumvent the limitations related to germline deletion, the authors have attempted to specifically delete p57 in MuSCs using the Pax7-CreERT2 driver. This approach has been proven unsuccessful in ablating p57 with high frequency. This is a technical issue that should be rectified. Without deleting p57 in the majority of MuSCs, in vivo experiments aimed at evaluating the role of p57 in homeostatic as well as regenerative conditions are simply not possible and the major tenet of this study, i.e., the function of p57 in adult MuSCs, cannot be conclusively determined.

3) Because of the low recombination of the p57 allele, in vitro experiments were presumably conducted with rare recombined MuSCs. Were there a selection process, the results may be difficult to interpret.

4) Based on the results presented in Figure 2, the authors conclude that p57 counteracts self-renewal. It remains unclear whether Pax7+ p57-null MuSCs proliferate or return to quiescence 30 days after injury. For this, it is necessary to evaluate cell proliferation by EdU labeling in vivo at later stages after tissue repair (30 days), as well as to determine cell anatomical position and expression of quiescent genes.

5) The cytoplasmic-nuclear shuttling of p57 should be documented upon MuScs in vivo activation. P57 subcellular localization could be determined in activated MuSCs on cultured myofibers with analysis of p57 protein localization upon in vivo activation. Analysis could be done on muscle stem cells at different early time points after tissue injury, either in tissues or on fixed cells after FACS isolation.

6) The fate of cells in which p57 has been overexpressed should be investigated to determine whether they are differentiating (Myogenin staining). Moreover, p57 overexpression experiments should be conducted in p57-null MuSCs.

7) No mechanistic insight of how p57 regulate different function in the cytoplasm and the nucleus is offered. P57 has been reported to associate with MyoD. Does nuclear p57 colocalize with MyoD at muscle regulatory regions? Even data that helps to exclude certain mechanisms would be useful in determining the subcellular functional role of p57.

We are also including the full reviews below in case the additional comments are helpful.

Reviewer #1:

The authors of this manuscript report that the cyclin-dependent kinase inhibitor p57 regulates skeletal muscle cell differentiation. Constitutive as well as conditional p57 knock-outs display increased proliferating muscle stem cells (MuSCs) and decreased differentiating muscle cells following muscle injury and regeneration. P57 was not expressed in quiescent MuSCs and activated in MuSCs entering the cell cycle. During cell differentiation p57 relocated from the cytoplasmic compartment to the nucleus where it is required to induce growth arrest.

Overall, the experiments are well conducted and the conclusions supported by the results.

While p57 has not been previously ablated in adult muscle cells, there have been several independent reports indicating that p57 either along with p21 or on its own regulates skeletal muscle cell proliferation and differentiation (Zhang et al., 1999; Naito et al., 2016).

Suggestions:

1) *Pax7^CreERT2^*; p57+/-; *ROSA^mTmG/+^* mice should be challenged with muscle injury and MuSC-mediated repair investigated.

2) The nuclear role of p57 in promoting growth arrest should be experimentally addressed.

P57 has been reported to associate with MyoD (Reynaud et al., 1999). The authors could evaluate whether p57 is recruited on selected MyoD targets in differentiating myocytes.

Reviewer #2:

In the manuscript entitled 'Cellular localization of the cell cycle inhibitor p57kip2 controls proliferation and growth arrest of adult skeletal muscle stem cells' the authors are investigating the role of the cell cycle inhibitor p57kip2 during adult skeletal myogenesis for the first time. p57kip2 has been previously studied as a regulator of stem cell quiescence in other stem cell system such as HSC and neural stem cells (Matsumoto et al., 2011; Zou et al., 2011;Furutachi et al., 2013). Previous work in myogenesis has shown that p57kip2 inhibits proliferation and controls muscle differentiation in embryonic muscle or in myoblasts cell line (C2C12) in vitro.

The manuscript by Mademtzoglou et al., attempts to demonstrate for the first time a role for p57kip2 during proliferation and differentiation of adult skeletal muscle satellite cells (MuSC). In particular, the authors showed that p57kip2 protein is not expressed in quiescent muscle SCs, but its expression is increased upon MuSC activation and retained during differentiation. By using genetic mouse model and ex vivo experiments they showed that p57kip2 is required for proper myogenic lineage progression. Genetic ablation of p57kip2 lead to increased proliferation and self-renew, but muscle differentiation is impaired. Interestingly, they showed that p57kip2 subcellular localization is progressively shifting from cytoplasm during activation to nuclear translocation upon differentiation.

There are two main issues with this manuscript:

1A) The authors revealed a role for p57kip2 during post-natal muscle growth and adult skeletal muscle regeneration relying on the germline p57kip2 mutant mice. However, compensatory upregulation of other cell cycle inhibitors from the Cip/Kip family (p18, p27), as shown previously in HSC (Zou et al., 2011), may diminish the effect of p57 in MuSC. An inducible knockout may give a more profound phenotype than that observed in straight KO mice, as suggested from severe MuSC differentiation defect derived from *Pax7^CreERT2^;p57^flox^* mice (Figure 3J-L) compared to more modest reduced differentiation from p57 mutant myoblasts, shown in Figure 3—figure supplement 1C. In addition, any proliferative phenotype during embryogenesis may negatively impact the adult SC in a regenerative context.

1B) In an attempt to use inducible mutant mice, the authors make a cursory comment that they were not able to achieve high efficiency recombination. This is a technical problem that should be rectified. The analysis of myogenic cell fate should include staining for Pax7, Ki67 and MyoD, at different time points after regeneration (d4, d7 and d30). These additional immunostains would further strengthen the conclusion that loss of p57 results in expansion of progenitors and increased self-renew at expense of differentiation.

1C) in vitro experiments were performed with rare recombined cells, this may be problematic if there is a selection process. Due to these potential artefacts, it is important the authors perform the above experiment.

2A) Despite their attempt to demonstrate that subcellular localization of p57kip2 directly affects the cycling status of myoblasts, they did not fully characterize the molecular mechanisms underlying p57kip2's cytoplasm-nuclear shuttling.

In particular, the authors failed to 1) Demonstrate the cellular phenotype of p57kip2 localization specifically in Pax7+ MuSCs and their progeny; 2) Fully understand the mechanism behind p57kip2 cytoplasm-nuclear shuttling and cell cycle/myogenic state.

2B) The authors generated retroviruses encoding the full length p57 (p57FL) or p57 lacking the nuclear localization signal (p57NLS). They showed that overexpression of p57-FL correlates with nuclear localization of the p57 protein and growth arrest based on the absence of Ki67 staining. NO phenotype with P57 lacking NLS. The authors should assess what is the fate of the arrested cells by immunostaining. Are they differentiating (Myogenin)?

In this experimental design, the authors exploit an overexpression system. The authors should perform retrovirus studies in the null background. Finally, can the authors use P57NLS to drive nuclear expression only?

2C) The manuscript lacks mechanistic detail of how P57 functions in a spatially- dependent manner. The authors speculate in the discussion to four potential mechanisms. It is important that some attempt is made to provide the readers with some mechanistic insight-even data that helps to exclude certain mechanisms that may be more tractable, would support the functional role of P57 localization.

Reviewer #3:

The present work by Mademtzoglou et al. investigates the role of the cell cycle inhibitor p57 on muscle stem cell function. The authors have previously reported that p57 deletion during embryonic development leads to defects in muscle progenitors' cell fate decisions. Here they extend these findings to adult muscle stem cells by showing that in a mixed background a small percentage of p57 knockout mice survived to adulthood and exhibited reduced body weight and myofiber cross-sectional area and increased fibrosis. These mice also exhibited a slight increase in muscle stem cell number, which on myofibers ex vivo or 30 days after tissue injury in vivo are more biased towards self-renewal at the expenses of myogenic commitment. They complement these studies by conditionally inducing p57 deletion in adult muscle stem cells, and upon isolation and culture they observed consistent results. They report that p57 is initially localized in the cytoplasm during early activation of muscle stem cells, and its progressive nuclear localization correlates with growth arrest and myogenic differentiation. Finally, overexpression of a full length, but not a mutant p57 lacking its nuclear localization signal, has a negative effect on myoblast proliferation. They conclude that p57 does not regulate muscle stem cell quiescence but rather their self-renewal vs. commitment cell fate choices.

Overall, the findings in this well written manuscript are interesting and novel, particularly the subcellular localization of p57 in muscle stem cells, although they could be strengthened by increasing the depth of the mechanism underlying the regulation of this process. Also, additional experiments are required in order to fully support the authors' interpretation.

1) Increased self-renewal or failure to reentry into quiescence: In Figure 2 the authors show that p57 knockout mice exhibit a mild increase in the number of muscle stem cells 30 days after injury, but no effect on MyoD expression. The authors conclude that p57 counteracts self-renewal. Are these cells proliferating or they efficiently went back to quiescence? Considering this gene has been previously implicated in stem cell quiescence, it might play a role here in promoting the reentry to this state after tissue repair, even if it is not required for the developmental entry in quiescence. The authors should control for proliferation by EdU labeling in vivo at later stage after tissue repair (30 days), as well as their anatomical location and expression of quiescent genes. In addition, analysis of muscle stem cells should be performed not only at 30 days but also at earlier stages after tissue injury, by immunostaining for Pax7, MyoD and EdU in vivo, in order to evaluate the dynamics of activation and commitment in the native tissue. This would strengthen the interpretation of the findings.

2) Efficiency of p57 floxing in muscle stem cells: The efficiency of the floxing in Figure 3 should be indicated, in order to evaluate whether the FACS isolation of labeled cells yields sufficient numbers to be representative of the stem cell pool, or a very low percentage that may hold an intrinsic bias in population heterogeneity. This should be discussed in the text.

3) Validating p57 expression dynamics upon in vivo activation: Data could be strengthened by reinforcing the observation of p57 subcellular localization in activated muscle stem cells on cultured myofibers, with analysis of p57 protein localization upon in vivo activation, i.e. analysis could be done on muscle stem cells at different early time points after tissue injury, either in tissues or immediately fixed after FACS isolation.

4) Effect of p57 loss on myogenic commitment: In Figure 5 the authors show that overexpression of a full length, but not a mutant p57 lacking its nuclear localization signal, has a negative effect on myoblast proliferation. It would be interesting to test whether there is a similar effect also on myogenic commitment.

[Editors' note: further revisions were requested prior to acceptance, as described below.]

Thank you for resubmitting your work entitled "Cellular localization of the cell cycle inhibitor Cdkn1c controls growth arrest of adult skeletal muscle stem cells" for further consideration at *eLife*. Your revised article has been favorably evaluated by Didier Stainier (Senior Editor), and a Reviewing Editor and the original reviewers, one of whom is a member of our Board of Reviewing Editors.

The manuscript has been improved but there are some remaining issues that need to be addressed before acceptance, as outlined below:

The authors have satisfactorily addressed most reviewers' comments and as a result the revised manuscript is substantially strengthened. There are some remaining concerns, detailed here below, that the authors should address.

1) Figure 3. Cdkn1c expression is abolished in ~ 100% of the Cdkn1c KO myoblast cultures (Figure 3B-D) and all the subsequent experiments reported in Figure 3 were conducted with these cultures. However, the myoblasts appear to be derived from MuSCs FACS-selected for Cdkn1c deletion. The authors should report the overall deletion efficiency (percentage of Cdkn1c-deleted MuSCs in *Pax7^CreERT2^; Cdkn1c^Flox^*), as this information is relevant to the interpretation of subsequent experiments.

2) New Figure 4. The authors have chosen to employ a *Pax7^CreERT2^* strain expressing only one Pax7 allele. As such, *Pax7^CreERT2^* mice have reduced Pax7 cells and display increased cell infiltration and reduced regenerating myofibers upon muscle injury. Further regeneration worsening in *Pax7^CreERT2^; Cdkn1c^Flox^* is thus to be ascribed to the compounding effect of inactivating both Pax7 and Cdkn1c, with Pax7 inactivation being the major contributor of reduced Pax7+cells (Figure 4H).

3) New Figure 4. It is possible that the number of MuSCs is not decreased but rather Pax7 at the level of the gene-protein is decreased. It is important to quantify MuSCs number using a different antibody (syn-4, or m-cadherin). The answer to this question would help to determine the severity of the Pax7 haplo-insufficiency phenotype. In addition, the images representing the figure need to be accompanied by quantification, for instance, the number and size of regenerating fibers across the 3 groups.

4) New Figure 4. The observed reduced number of Pax7+ cells should also be discussed in more detail, as in most of the other assays the authors show that deletion of Cdkn1c leads to increased self-renewal and impaired myogenic differentiation. It would be useful if the authors could comment on this phenotypic difference in the different conditions. A more thorough analysis of the in vivo phenotype may resolve these contradictions.

---

## [Author Response]

Essential revisions:1) Beside causing perinatal lethality, germline p57 ablation induces gross developmental defects owing to its role in controlling quiescence of hematopoietic and neural stem cells. Thus, the combination of very few (~4%) p57 knock-out surviving animals and the extensive non-cell specific function of p57 make the results of the presented experiments difficult to interpret.

In the revised manuscript, we added data on satellite-cell-specific p57 deletion in adult MuSCs, using the tamoxifen-inducible Pax7-CreERT2 recombinase. These data are presented in (new) Figure 4. Inclusion of this figure changed the numbering of subsequent figures and figure supplements.

2) To circumvent the limitations related to germline deletion, the authors have attempted to specifically delete p57 in MuSCs using the Pax7-CreERT2 driver. This approach has been proven unsuccessful in ablating p57 with high frequency. This is a technical issue that should be rectified. Without deleting p57 in the majority of MuSCs, in vivo experiments aimed at evaluating the role of p57 in homeostatic as well as regenerative conditions are simply not possible and the major tenet of this study, i.e., the function of p57 in adult MuSCs, cannot be conclusively determined.

In order to provide more conclusive evidence of p57 function in adult MuSCs, we treated *Pax7^CreERT2^; p57^flox^* animals with tamoxifen diet for a total of five weeks. This approach allowed the ablation of p57 in adult muscle satellite cells and their post-regeneration progeny, in contrast to our previous approaches of 3-5 i.p. tamoxifen injections or 1-2 weeks of tamoxifen diet. Ablation of p57 in MuSC led to impaired muscle repair after injury. We also provide relevant controls. The new data are included in new Figure 4.

*3) Because of the low recombination of the p57 allele,* in vitro *experiments were presumably conducted with rare recombined MuSCs. Were there a selection process, the results may be difficult to interpret.*

We agree with the reviewers remark and in the revised manuscript, we strengthened these in vitroresults by in vivoanalysis of MuSC-specific *p57* ablation (in *Pax7^CreERT2^; p57^flox^* animals) during muscle repair (Figure 4). Under these conditions, recombined and non-recombined cells were equally challenged by cardiotoxin-induced muscle degeneration and regeneration and showed a severely compromised potential to reconstitute normal muscle tissue.

Of note, we conducted all in vitroexperiments with freshly isolated 70-80% (recombined) MuSCs, avoiding passages that potentially increase the presumed selection process.

4) Based on the results presented in Figure 2, the authors conclude that p57 counteracts self-renewal. It remains unclear whether Pax7+ p57-null MuSCs proliferate or return to quiescence 30 days after injury. For this, it is necessary to evaluate cell proliferation by EdU labeling in vivo at later stages after tissue repair (30 days), as well as to determine cell anatomical position and expression of quiescent genes.

In our new data shown in Figure 2—figure supplement 1D, PAX7+ *p57*-null MuSCs located underneath the LAMININ+ basal lamina returned to quiescent state 30 days after injury confirmed by EdU staining.

5) The cytoplasmic-nuclear shuttling of p57 should be documented upon MuScs in vivo activation. P57 subcellular localization could be determined in activated MuSCs on cultured myofibers with analysis of p57 protein localization upon in vivo activation. Analysis could be done on muscle stem cells at different early time points after tissue injury, either in tissues or on fixed cells after FACS isolation.

We performed muscle injury and monitored p57 subcellular presence at early and late time points following regeneration in muscle sections. We observed cytoplasmic p57 at D3 post-injury, while later on p57 was present in the central nuclei of the newly-formed, regenerated myofibers. We present data on D3 and D13 postinjury in updated Figure 5.

6) The fate of cells in which p57 has been overexpressed should be investigated to determine whether they are differentiating (Myogenin staining). Moreover, p57 overexpression experiments should be conducted in p57-null MuSCs.

In the revised manuscript we analyzed the effects of p57 overexpression on differentiation, by quantifying the Myogenin+ cells at early and late time points of Myogenin expression in single myofiber culture. Due to the lack of significant difference, we further investigated whether p57 overexpression acts earlier during myogenic differentiation, by analyzing the MyoD+ cells after 72h of single myofiber culture. At that time point, MyoD labels not only activated myoblasts but also those that will proceed to differentiation. Furthermore, we performed overexpression experiments in p57-deficient MuSCs after FACS isolation as requested. These data are now shown in new Figure 6—figure supplement 2.

7) No mechanistic insight of how p57 regulate different function in the cytoplasm and the nucleus is offered. P57 has been reported to associate with MyoD. Does nuclear p57 colocalize with MyoD at muscle regulatory regions? Even data that helps to exclude certain mechanisms would be useful in determining the subcellular functional role of p57.

We thank the reviewers for this comment. To gain insight into the molecular mechanisms of p57 action, we used data generated by a MyoD ChIP sequencing experiment (Cao et al., 2010) to identify MyoD binding sites at muscle regulatory regions. We then performed ChIP experiments with an anti-p57 antibody to test whether p57 shares these binding sites with MyoD. The results are shown in new Figure 6—figure supplement 1.

Cao Y, Yao Z, Sarkar D, Lawrence M, Sanchez GJ, Parker MH, MacQuarrie KL, Davison J, Morgan MT, Ruzzo WL, Gentleman RC, Tapscott SJ. (2010). Genome-wide MyoD binding in skeletal muscle cells: a potential for broad cellular reprogramming. Dev Cell 18 (4): 662-674.

[Editors' note: further revisions were requested prior to acceptance, as described below.]

The manuscript has been improved but there are some remaining issues that need to be addressed before acceptance, as outlined below:The authors have satisfactorily addressed most reviewers' comments and as a result the revised manuscript is substantially strengthened. There are some remaining concerns, detailed here below, that the authors should address.1) Figure 3. Cdkn1c expression is abolished in ~ 100% of the Cdkn1c KO myoblast cultures (Figure 3B-D) and all the subsequent experiments reported in Figure 3 were conducted with these cultures. However, the myoblasts appear to be derived from MuSCs FACS-selected for Cdkn1c deletion. The authors should report the overall deletion efficiency (percentage of Cdkn1c-deleted MuSCs in Pax7^CreERT2^; Cdkn1c^Flox^), as this information is relevant to the interpretation of subsequent experiments.

We added data on the recombination efficiency in Figure 3—figure supplement 1. The lack of Cdkn1c expression in quiescent MuSCs prevented us from evaluating the percentage of *Cdkn1c*‐deleted MuSCs in *Pax7^CreERT2^;Cdkn1c^Flox^* mice. Instead, we used *Pax7^CreERT2^;Rosa^mTmG^* mice, which express GFP as a result of recombination, and we found that one-week of tamoxifen administration leads to 84.5% recombination (percentage of Pax7+GFP+ cells among Pax7+ satellite cells).

Furthermore, we quantified the recombination efficiency for the experiments presented in Figure 4 and added these data in Figure 4—figure supplement 1 of the revised manuscript.

2) New Figure 4. The authors have chosen to employ a Pax7^CreERT2^ strain expressing only one Pax7 allele. As such, Pax7^CreERT2^ mice have reduced Pax7 cells and display increased cell infiltration and reduced regenerating myofibers upon muscle injury. Further regeneration worsening in Pax7^CreERT2^; Cdkn1c^Flox^ is thus to be ascribed to the compounding effect of inactivating both Pax7 and Cdkn1c, with Pax7 inactivation being the major contributor of reduced Pax7+cells (Figure 4H).

We agree with the reviewers' remark and we added relevant discussion in the revised manuscript.

3) New Figure 4. It is possible that the number of MuSCs is not decreased but rather Pax7 at the level of the gene-protein is decreased. It is important to quantify MuSCs number using a different antibody (syn-4, or m-cadherin). The answer to this question would help to determine the severity of the Pax7 haplo-insufficiency phenotype. In addition, the images representing the figure need to be accompanied by quantification, for instance, the number and size of regenerating fibers across the 3 groups.

When quantified using M-cadherin, MuSCs were reduced from wt to Cre control and further from Cre control to *Cdkn1c* cKO (Author response image 1), as originally reported in Figure 4.

**Author response image 1. respfig1:** Quantification of M-cadherin+ MuSCs in regenerating muscle of wildtype littermate (Wt; *Pax7^+^;Cdkn1c^+^*), Cre control (*Pax7^CreERT2^*), and *Cdkn1c* cKO (*Pax7^CreERT2^;Cdkn1c^Flox^*) mice at D7 post-cardiotoxin injection.

The post-injury fiber size distributions were added in the revised manuscript, in Figure 4—figure supplement 1.

4) New Figure 4. The observed reduced number of Pax7+ cells should also be discussed in more detail, as in most of the other assays the authors show that deletion of Cdkn1c leads to increased self-renewal and impaired myogenic differentiation. It would be useful if the authors could comment on this phenotypic difference in the different conditions. A more thorough analysis of the in vivo phenotype may resolve these contradictions.

The difference in MuSC self-renewal (*Cdkn1c* mutant) or exhaustion (*Cdkn1c* cKO) could be attributed to the constitutive versus conditional deletion of the gene, respectively, that could activate different compensatory mechanisms. The myogenic differentiation defect is consistent in both mutant and cKO mice, in our in vivo or ex vivo analyses. These clarifications have been added to the text and the phenotypes of germline and conditional KO mice are comparatively discussed in the updated manuscript.